

# Extreme Mediterranean cyclones and associated variables in an atmosphere-only vs an ocean-coupled regional model

Marco Chericoni[1,2], Giorgia Fosser[1], Emmanouil Flaounas[3], Gianmaria Sannino[2,4], Alessandro Anav[2]

[1]University School for Advanced Studies IUSS, Pavia, 27100, Italy
[2]Italian National Agency for New Technologies, Energy and the Environment (ENEA), Rome, 00196, Italy
[3]Institute of Oceanography, Hellenic Centre for Marine Research, Athens, 19013, Greece
[4]ICSC Italian Research Center on High-Performance Computing, Big Data and Quantum Computing, Bologna, 40033, Italy

*Correspondence to*: Marco Chericoni (marco.chericoni@iusspavia.it)

**Abstract.** Complex air-sea interactions play a major role in both the variability and the extremes of the Mediterranean climate. This study investigates the differences between an atmosphere-only and an ocean-coupled model in reproducing Mediterranean cyclones and their associated atmospheric fields. To this end, two simulations are performed using the ENEA-REG regional Earth system model at 12 km atmospheric horizontal resolution over the Med-CORDEX domain, both driven by ERA5 reanalysis, for a common 33-year period (1982–2014). The atmosphere stand-alone simulation uses the WRF model with prescribed ERA5 Sea Surface Temperature (SST), while in the second WRF is coupled to the MITgcm ocean model at horizontal resolution of 1/12°. A cyclone track method, based on sea level pressure, is applied to both simulations and to the ERA5 reanalysis to assess the model capability to reproduce the climatology of intense, potentially most impactful, cyclones. Results show that the seasonal and spatial distribution of the 500 most intense cyclones is similarly reproduced between WRF and ERA5, regardless the use of the coupling. The two simulations are then compared in terms of sub-daily fields at the cyclones' maximum intensity. Differences in SST distribution between the models primarily control variations in atmospheric variables, not only at the surface, but throughout the planet boundary layer, due to the mixing of the turbulent processes, enhanced during intense cyclones. Additionally, the research investigates the cyclone effects on ocean properties in the coupled simulation, revealing that strong winds enhance surface heat fluxes and upper ocean mixing, while lowering SST. The analysis shows the effectiveness of the coupled model in representing dynamic and thermodynamic processes associated with extreme cyclones across both the atmosphere and the ocean.

## 1 Introduction

The Mediterranean region is intriguing to climate scientists because is a hot spot for climate change and presents unique climatic features, that arises from a complex morphology and land-sea distribution and its position bridging the tropics and mid-latitudes (Tuel and Eltahir, 2020). Besides, the ocean and atmosphere interact at the air-sea interface, across a wide range of spatial and temporal scales, generating strong air-sea feedback. From one side, the large-scale atmospheric dynamics





influence ocean variability (Gill 2016), with strong winds enhancing surface heat fluxes and upper ocean mixing, while lowering Sea Surface Temperature (SST). On the other side, ocean structures at meso-scale impact atmospheric dynamics (Chelton et al., 2001), affecting air temperature, frictional stress, surface wind patterns and atmospheric boundary layer stability, thus significantly influencing the water cycle (Cassola et al., 2016; Chelton et al., 2004; Meroni et al., 2018; Senatore et al., 2020; Small et al., 2008). These small-scale air-sea feedback processes interact with large scale structures, such as mid-latitude cyclones, entering the Mediterranean basin from the Atlantic region. Mediterranean cyclones are typically less intense, smaller, and shorter-lived compared to both tropical and mid-latitude cyclones forming over open oceans. However, their formation is very common, making the Mediterranean basin one of the regions with the highest occurrence of cyclones in the world (Neu et al., 2013; Ulbrich et al., 2009). Despite their relative weakness, Mediterranean cyclones often bring extreme precipitation and strong winds, especially in winter and, in some cases, in autumn, causing significant socio-economic and environmental impacts, particularly in densely populated coastal areas. Thus, a deeper understanding and a more realistic representation of air-sea interaction processes during cyclones is crucial from an impact perspective.

International initiatives like the Mediterranean Experiment (MEDEX, 2000-2010; Jansa et al. 2014) and the Hydrological Cycle in the Mediterranean Experiment (HyMeX, 2010-2020; Drobinski et al. 2014) have contributed to our understanding of cyclone dynamics, as well as their impacts on the Mediterranean water cycle through coordinated community efforts. Multiple studies indicate that cyclones in the Mediterranean region account for at least 70% of extreme rainfall events (Catto and Pfahl, 2013; Jansa et al., 2001; Nissen et al., 2013; Pfahl et al., 2014; Pfahl and Wernli, 2012), with deep convection and warm conveyor belt processes being the main contributors to heavy rainfall (Flaounas et al., 2018c, 2019). Additionally, these cyclones are responsible for the majority of extreme wind storms (Hewson and Neu, 2015; Nissen et al., 2010, 2014) and for the formation of high-impact weather events (Llasat et al., 2010, 2013). Those events produce a high variability in the evaporation and precipitation fields, playing a significant role in the Mediterranean Sea water budget (Flaounas et al., 2016; Romanski et al., 2012). Climatological studies show that the most intense cyclones occur predominantly in winter, forming over the leeward side of the Alps and reaching their sea level pressure (SLP) minima over the sea (Campins et al., 2011; Flaounas et al., 2015; Flocas et al., 2010; Trigo et al., 2002). They develop within a baroclinic atmosphere, influenced by upper-tropospheric precursors, primarily in the form of narrow potential vorticity (PV) streamers that intrude into the Mediterranean region (Raveh-Rubin and Flaounas, 2017). Diabatically generated PV at middle and lower atmospheric levels also impact cyclone development, with latent heat release as the primary source of PV reinforcement from ground level to the mid-troposphere, strengthening cyclonic circulations (Fita et al., 2006). Other local factors, like orographic effects and air-sea interactions, play an important, but secondary role (Campins et al. 2000; Trigo et al. 2002; Buzzi et al. 2003; Horvath et al. 2006, 2008; McTaggart-Cowan et al. 2010).

Given cyclones' significant impact on Mediterranean climate, it is crucial for models to accurately reproduce their dynamics to assess climate impacts on human and natural environments. Regional Climate Models (RCMs) have been long ago employed



to analyse climate dynamics across different spatial scales and several recent studies demonstrated their benefits in reproducing
Mediterranean cyclones (Calmanti et al., 2015; D'Onofrio et al., 2014; Flaounas et al., 2013; Guyennon et al., 2013). However,
RCMs performance often depend on the quality of the coarse resolution SST used as lower boundary conditions, that becomes
even more challenging in the climate change context when reanalysis datasets are not available. Thus, integrating regional
atmosphere and ocean model components into a coupled system is being increasingly challenged by research groups and
operational centres (Gentile et al., 2022; Lewis et al., 2018; Ricchi et al., 2017; Varlas et al., 2018; Wahle et al., 2017). In
particular, over the Mediterranean region, the coupled atmosphere-ocean RCMs, within the Med-CORDEX community (Ruti
et al., 2016), offer an opportunity to investigate the impact of increased resolution and air-sea coupling on extreme events,
such as intense Mediterranean cyclones. Directly simulating the effect of the dynamical ocean state on atmospheric surface
processes is expected to better simulate surface fluxes, leading to improved representation of weather systems characterised
by strong near-surface wind speeds, such as in extratropical cyclones. Previous studies indicate that coupling atmosphere and
ocean over the Mediterranean affects simulated 2 m temperature, evaporation, precipitation and wind speed, as well as the
Mediterranean water budget (e.g., Van Pham et al. 2014; Lebeaupin Brossier et al. 2015; Ho-Hagemann et al. 2017), with
high-resolution coupled models enhancing the representation of sea surface fluxes (Artale et al., 2010; Dubois et al., 2012;
Gualdi et al., 2013; Somot et al., 2008). Berthou et al. (2014, 2015, 2016) found that only a minor part of the change in
precipitation was due to proper air-sea coupling effects, while the long-term difference in SST between the simulations were
responsible for most of the change. In terms of Mediterranean cyclones, Flaounas et al. (2018a) found that the most intense
are similarly reproduced in both coupled and uncoupled RCMs, suggesting that the coupling system has a limited effect on the
climatology and intensity of the cyclones, primarily because the cyclogenesis is mainly driven by upper tropospheric forcing.
However, the weak impact of air-sea interactions may also be attributed to the coarse resolution of the used RCMs, ranging
from 20 to 50 km. For example, Akhtar et al. (2014) demonstrated, based on selected case studies, that the coupling effect on
medicanes, i.e., Mediterranean tropical-like cyclones (Miglietta, 2019), becomes significant for model resolutions at around
10 km. They also showed that at higher resolutions, the coupled model improves the track length, core temperature, and wind
speed of simulated medicanes compared to atmosphere-only simulations, thanks to better resolved mesoscale processes and
turbulent fluxes. However, it is unknown if these findings can be confirmed at climatological scale.


All the studies on the impacts of atmosphere-ocean coupling on the Mediterranean climate variability and extremes examined
only the atmosphere, while the coupling potentially allows the ocean to respond to an atmospheric forcing, such as a cyclone.
Moreover, previous research at climatological scale focuses only on surface variables (Artale et al., 2010; Dubois et al., 2012;
Gualdi et al., 2013; Somot et al., 2008), while is still an open question to which extent in the vertical column a different SST
can influence the atmospheric state, especially during extreme cyclones events, when the vertical turbulent processes are
expected to be stronger. Thus, this research aims to fill these knowledge gaps investigating how Mediterranean cyclones affects
simultaneously the atmosphere and the ocean at different vertical levels. Comparing a high-resolution atmosphere-ocean





coupled RCM and its atmospheric stand-alone version, this study seeks to bring new insights on how the energy redistributes in the entire atmosphere-ocean system, during extreme cyclone events.

The specific questions that are addressed in this paper are:

1. Does the high-resolution atmosphere-ocean coupled RCM better represent the climatology of extreme Mediterranean cyclones?

2. To which extent in the vertical column, and through which physical mechanisms, the explicitly resolved SST distribution and sea surface fluxes impact the precipitation, and the wind speed during extreme cyclones?

3. Does the coupling allow for the depiction of the ocean response to the extreme cyclones?

For a more comprehensive analysis, two seasons are considered: the winter when the cyclones are more intense and autumn when the role of the SST and the air-sea fluxes on extreme events is expected to be stronger.

The present paper is structured as follows: next session describes the model and the methods employed. Section 3 addresses the research questions, focusing on the cyclone climatology, on the cyclones' sub-daily fields and on their impact on both the

atmosphere and ocean structures. Finally, section 4 summarises the findings of the work and presents the concluding remarks.

## 2 Models and methodology

### 2.1 Model description

To assess the impact of high-resolution atmosphere-ocean coupling on the dynamics and thermodynamics of extreme cyclones, two hindcast RCM simulations are performed. The first simulation, referred to as STD, uses the mesoscale Weather Research

and Forecasting model (WRF version 4.2.2) with prescribed SST from ERA5 reanalysis (Hersbach et al., 2020). The second simulation (henceforward CPL) uses the ENEA-REG regional Earth system model (Anav et al., 2021) where WRF has the same set-up and physical parametrizations than STD, but is coupled to the Massachusetts Institute of Technology General Circulation Model (MITgcm version c65; (Marshall et al. 1997), extensively used in recent studies to investigate the Mediterranean circulation at different resolutions and time-scales (e.g. Palma et al. 2020; Sannino et al. 2022). Thus, the only

difference between the STD and the CPL simulation resides in the SST over the Mediterranean Sea, which derives from the ERA5 SST reanalysis (daily, $\Delta x = 0.25°$) in STD, whereas it comes interactively from MITgcm (3-hourly, $\Delta x \simeq 1/12°$) in CPL. The WRF horizontal resolution is 12 km, while the ocean component of the CPL has a resolution of $1/12°$ (approximately 10 km). The two simulations initialised and forced by ERA5 (Hersbach et al., 2020) and ORAS5 (Zuo et al., 2019) reanalysis, respectively for the atmospheric and ocean components, cover the Med-CORDEX region (Anav et al., 2024) over the period

1980-2014. The first two years are used as a spin-up period and thus the analysis is performed for the period 1982-2014.

The ENEA-REG (Anav et al. 2024) is a regional Earth system model designed for high-resolution climate studies. It includes multiple components: the atmosphere, ocean, land, and river routing. Data exchange and interpolation among these components are managed using the RegESM coupler, as described by Turuncoglu (2019). RegESM is based on the Earth System Modeling Framework (ESMF) version 7.1 and uses the NUOPC (National Unified Operational Prediction Capability)





layer for interconnections, synchronization, and horizontal grid interpolation. ENEA-REG incorporates the Weather Research and Forecasting model (WRF version 4.2.2) for atmospheric dynamics, the Noah-MP, embedded within WRF, for the land scheme, the Massachusetts Institute of Technology General Circulation Model (MITgcm version z67; Marshall et al. 1997) for ocean state and circulation, and the Hydrological Discharge model (HD version 1.0.2, Hagemann and Gates 2001) for simulating freshwater fluxes over the land surface and river discharge to the ocean model. A key improvement in the ocean

model is the addition of a full non-linear free-surface formulation (Campin et al., 2004). The ocean boundary conditions are provided as monthly sea level fields.

The atmospheric and ocean models exchange sea surface temperature (SST), surface pressure, wind components, freshwater (evaporation-precipitation), and heat fluxes. Net heat flux is computed from net longwave and shortwave radiation, latent heat, and sensible heat fluxes, with shortwave radiation penetrating the ocean as a separate term. The hydrological model uses

surface and sub-surface runoff from WRF to compute river discharge, which it then exchanges with the ocean component to close the water cycle. The coupling time step between the ocean and atmosphere is 3 hours, while the hydrological model is coupled daily. For more details on model configuration and main physical parametrizations for the atmosphere and the ocean component, refer to Anav et al. (2024).

## 2.2 Methods

### 2.2.1 Cyclone tracking

A storm track method is applied to both ERA5 reanalysis and RCM simulations. This method is identical to the one used in Flaounas et al. (2023), called "CYCLOYTRACK", adapted from Flaounas et al. (2014), and uses Mean Sea Level Pressure (MSLP), at 6 hourly frequencies, to identify cyclone centres instead of relative vorticity at 850 hPa as in the original version (Flaounas et al., 2014). To identify cyclone centres, a Gaussian filter with a 150 km kernel and sigma value of 2 is first used

to smooth the MSLP input fields. Cyclone centres are thus identified as grid points with lower MSLP than their eight neighbour ones. Starting from the deepest cyclone centre, the algorithm constructs possible tracks by connecting centres across consecutive time steps within 250 km radius. Among the candidate tracks, the algorithm will eventually select the one with the least average MSLP difference. To note that WRF MSLP is upscaled to the grid of ERA5 before applying the storm tracking. This was done to assure a fair comparison of tracks between model and reanalysis (Kouroutzoglou et al., 2011), but

also to limit the detection of small and weak cyclonic features in WRF model outputs that typically have minimal influence on climate dynamics and extremes of the area (Flaounas et al., 2021). A terrain filter of 800 m altitude has been also applied to discard cyclone tracks over complex high mountain environments. This allows us first to focus on the intense cyclones over the sea and second to filter out algorithm artefacts, that tend to form in mountain volumes due to the extrapolation of pressure fields on sea level (Neu et al., 2013). Sensitivity tests were performed to evaluate the impact of the used height filter on the

number of detected cyclones, but no major differences were found among of 500, 800 and 1000 m.



Finally, only cyclones that present their minimum SLP tracking point within the area outlined by solid lines in Fig. 1 are considered in this study. Therein, the algorithm detects a total of 2805 cyclones in STD, 2695 in CPL and 2735 in ERA5. Among those, the 500 most intense cyclones have been retained (henceforward called extreme cyclones). Cyclones intensity is given by the minimum SLP that cyclone attains during its lifetime (i.e. duration of the track).

**2.2.2 Models comparison**

To compare CPL with STD in terms of sub-daily fields associated to the cyclones, the same events between the two simulations are selected, for a total of 341 cyclones from the 500 most intense. Two cyclones are considered the same event if their minimum of SLP is within a 500 km distance and within a time range of 12 hours. The comparison between STD and CPL is performed at their original spatial resolution of 12 km, and focuses on the mature stage of each cyclone, i.e. the three tracking

timesteps around the minimum SLP. To note that the WRF output frequency is 6 hours, thus the mature stage lasts from 6 hours before to 6 hours after the time of the minimum SLP tracking point. The analysis covers both winter and autumn, to cover the different state of the atmosphere and the ocean in these seasons. With these criteria, 140 in DJF and 59 in SON extreme cyclones are found in common between CPL and STD. Several atmospheric fields are analysed, namely temperature, precipitation, evaporation, wind speed, specific humidity, planet boundary layer (PBL) height and potential temperature

vertical gradient between 950 hPa and 1000 hPa (henceforward referred to as the Θ gradient), Eq. (1):

$$\nabla_z \theta = \frac{\theta_{950} - \theta_{1000}}{950 - 1000} \frac{K}{hPa} .\tag{1}$$

The fields are computed in each grid point of the investigation area (Fig. 1, dashed lines) during the mature stage of each cyclone, i.e. timestep of the minimum SLP plus the one before and after it, and then averaged over the number of cyclones in common between STD e CLP (i.e. 140 in DJF and 59 in SON). To note that precipitation (total and convective) and evaporation

are cumulated over the three timesteps considered, and not averaged as the other variables. These fields are referred to as "cyclone associated atmospheric fields", also called "cyclones composite fields". We acknowledge that this methodology includes in the average areas that are not affected by the cyclones. However, this would not significantly alter the results since the influence of cyclones on the atmospheric state is independent of the location of the cyclones in the Mediterranean Sea. On the other hand, our strategy allows to overcome the slight location mismatch between CPL and STD (i.e. linked with 500 km

maximum distance between the minimum of SLP) when computing the differences.

The atmospheric field differences between CPL and STD are normalised with the STD value for each grid cell, Eq. (2):

$$\Delta = \frac{100\,(CPL - STD)}{STD}\,\% ,\tag{2}$$

expected for the temperature and the Θ gradient differences. Note that the same timesteps (the mature stage of CPL cyclones) was used in both CPL and STD to compute the composite field differences.

The statistical significance of atmospheric field differences between STD and CPL during extreme winter cyclones are validated using a bootstrapping method (Efron et al. 1993). For this purpose, 1000 bootstrap surrogates are generated by





randomly selecting with replacement 140 cyclones from the list of the common extreme winter cyclones. Note that the same selection was used for both CPL and STD. The differences in the atmospheric fields between STD and CPL, calculated in each grid point for each bootstrap surrogate, are considered significant at the 5% level if the 2.5-97.5% confidence interval for the difference does not include the zero. In addition, the mean climatological winter SST distribution of both CPL and STD (ERA5 in this case) are validated against the Reprocessed Mediterranean dataset (MED-REP-L4, Pisano et al. 2016; Saha et al. 2018; Merchant et al. 2019;), which is a daily, satellite-based reconstruction of SST, with a spatial resolution of 0.05° available through the portal of the Copernicus Marine Service (CMEMS; https://marine.copernicus.eu/access-data). The statistical significance of the SST differences between the models and the MED-REP-L4 dataset are validated with the same methodology applied for the atmospheric fields. Hence, 1000 bootstrap surrogates are created by randomly selecting, with replacement, 99 winter months between 1982 and 2014. The SST differences, calculated at each grid point for each bootstrap surrogate, are considered significant at the 5% level if the 2.5-97.5% confidence interval for the difference does not include the zero. To investigate the connection between SST differences and atmospheric field differences during cyclone events, the Pearson correlation coefficients (R), and the p-values (for its significance) are computed for the grid points of differences that are statistically significant.

The last analysis focuses on the ocean component of the CPL model and aims to evaluate the impact of the cyclones on the ocean structures. For the CPL model, the vertical profile of the ocean temperature during the passage of extreme cyclones, both in DJF and in autumn (SON), is analysed and compared with the high-resolution Mediterranean Sea physical reanalysis (CMEMS MED-Currents; Escudier et al. 2020, 2021), developed in the Copernicus Marine Environment Monitoring Service framework. This reanalysis dataset is available from 1987 to now at 1/24° (ca. 4-5 km) grid resolution with 141 unevenly spaced vertical levels over the Mediterranean area. So, for this analysis, within the same extreme cyclones between CPL and STD, only those between 1987 and 2014 with their minimum SLP over the Mediterranean Sea, have been selected (102 cyclones in DJF and 43 in SON), and compared to CMEMS MED-Currents over the same events. The vertical profiles for both datasets are analysed 2 days before, during, and 2 days after the passage of the cyclone. The profiles represent the average over a circular area, with a radius of 1.5°, around the minimum SLP tracking point and over the cyclones considered. In addition, the temporal variation of the SST between 5 days before and 5 days after each cyclone in both DJF and SON is computed for CPL and STD and compared with CMEMS MED-Currents over the same events and circular area used in the analysis of the ocean vertical profile.



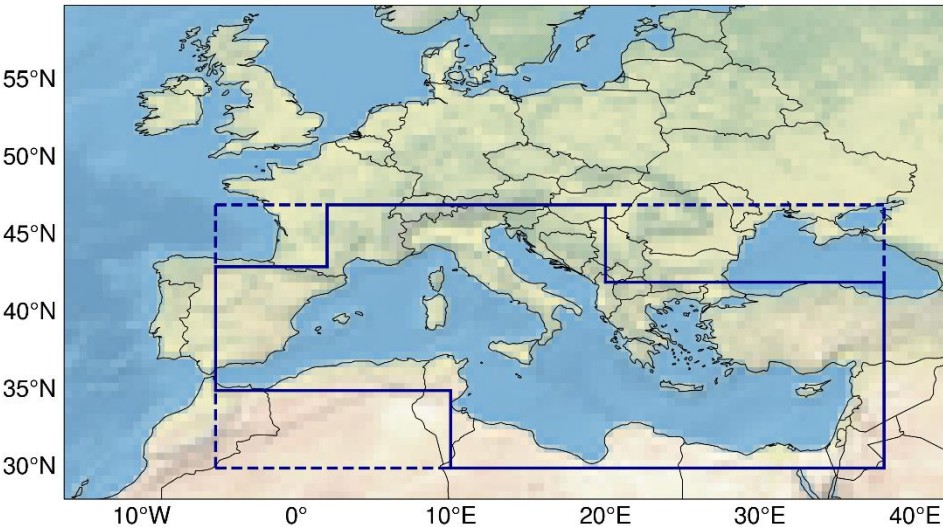

**Figure 1: Med-CORDEX domain. Cyclones are retained if their minimum SLP tracking point is present within the area outlined by solid lines. The atmospheric fields are computed within the rectangular area outlined by dashed lines.**

## 3 Results

### 3.1 Climatology of extreme Mediterranean cyclones

Figure 2 shows how the mean statistical properties (i.e., intensity, lifetime, and speed) and the seasonal cycle of the extreme cyclones are similarly reproduced between ERA5 and the two RCMs (i.e. STD and CPL) upscaled at ERA5 resolution (see section 2.2.1). The cyclones present a maximum intensity of 975 hPa, a mean lifetime of 4 days and a mean speed of 20 km h$^{-1}$. These results are in fair agreement with the most intense cyclones in ERA5 as detected by different cyclone tracking methods and in composite reference tracks for the Mediterranean (Flaounas et al., 2023). Also, the spatial distribution of the extreme cyclones is similarly reproduced by the models compared to ERA5 (Fig. 3), highlighting the capability of the RCMs to reproduce the climatology of Mediterranean cyclones. This can be expected since intense cyclones are mainly driven by large-scale upper-tropospheric forcings, potential vorticity streamers originating from the polar jet (Chaboureau et al., 2012; Fita et al., 2006; Flaounas et al., 2015, 2021; Neu et al., 2013; Raveh-Rubin and Flaounas, 2017), inherited for both simulations from ERA5 atmospheric lateral boundary conditions. However, cyclones seasonality and location also depend on fine scale atmospheric processes, such as convection, which are simulated in the models at a higher resolution than ERA5. Consequently, differences appear in both seasonal and spatial distribution due to the different native resolutions. In particular, a greater occurrence of summer and spring cyclones (Fig. 2d and Fig. S1 (supplementary)) is found compared to ERA5 and also over land (Fig. 3), likely due to RCMs' capacity to resolve more realistically the impact of orography on cyclone dynamics, resulting





in local deeper minima of SLP over complex terrain. In fact, the ERA5 cyclones are mainly concentrated over the Tyrrhenian

and Adriatic Sea (Fig. 3a), while the RCMs (Fig. 3a and b) presents higher frequency of cyclones over land and over Aegean

and Levantine Sea. Differences between STD and CPL are limited and non-significant, leading at the conclusion that the

atmosphere-ocean coupling has only a weak impact on the climatology and statistical properties of extreme Mediterranean

cyclones, as already found in Flaounas et al. (2018b).

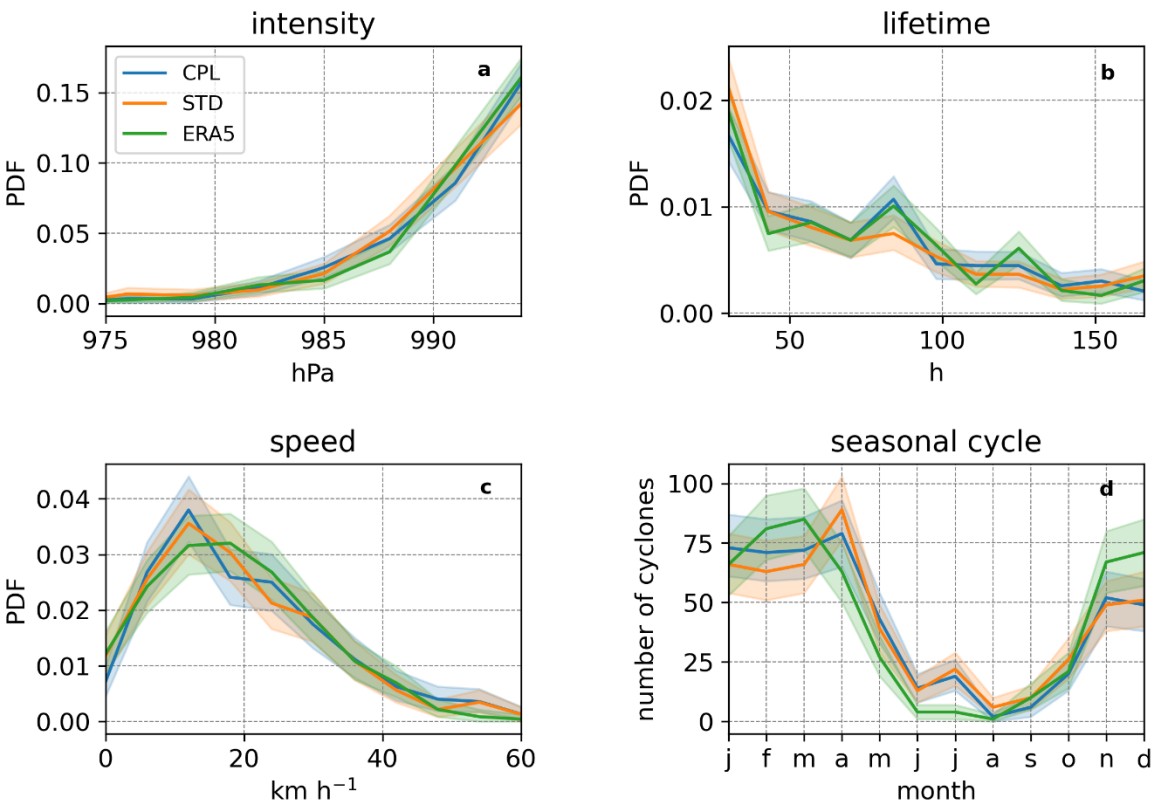

**Figure 2: Statistics, intensity (a), lifetime (b) and speed (c), and seasonal cycle (d) of the 500 most intense cyclones in STD, CPL and ERA5. The colour band represent the 2.5-97.5% confidence interval within the 1000 bootstrap surrogates.**





**Figure 3: Spatial distribution of the minima of the 500 most intense cyclones, called extreme cyclones, in ERA5 (top), STD (middle) and CPL (bottom). To note that to highlight the cyclones' area of influence, each cyclone is represented by a square area, 3 degrees each side, centred in the minimum SLP tracking point of its track.**





**3.2 Impact of the SST distribution on cyclones' precipitation and wind speed**

Heavy precipitation and strong wind speed, associated to cyclones, often lead to severe socio-economic and environmental impacts on the Mediterranean region, particularly in densely populated coastal areas. The following analysis evaluates the impact of the different SST distribution and surface fluxes between the CPL and the STD model on the atmospheric fields, during winter extreme cyclones in common between CPL and STD. Similar analysis is performed also for autumn season and figures can be found in supplementary. The analysis is structured as follows: firstly, the mean distribution of the atmospheric fields associated to extreme cyclones are presented for the STD simulation (section 3.2.1), then the SST distributions of CPL and STD are analysed and compared with observations (section 3.2.2), and finally the atmospheric differences between STD and CPL are investigated, focusing on the physical mechanism behind that (section 3.2.3).

**3.2.1 Atmospheric fields of STD during extreme cyclones**

During winter extreme cyclones, STD simulation shows that the precipitation predominantly accumulates over the coastal regions (Fig. 4a), especially over the eastern Adriatic and Ionian Sea, the western Turkish coast, and the Italian coast. This precipitation pattern is associated with winter cyclones generally coming from the west, as indicated by Flaounas et al. (2015) and Raveh-Rubin and Flaounas (2017). These cyclones interact with the complex orography of the basin and turning into precipitation when they reach the coast. The distribution of convective precipitation (Fig. 4b) is mainly concentrated over the sea, where the PBL is higher (Fig 4c), and close to the coastal regions where the sharp transition between sea and land fosters the convection processes. The wind blows mainly from the gulf of Lyon, where maximum speed is reached (above 14 m s$^{-1}$), to the north African coast and then deviates towards the Ionian Sea and Greece (Fig. 4d). A higher evaporation is found over the same area of high wind speed and reaches its maximum in the gulf of Lyon (Fig. 4e), due to the high wind speed that fosters the heat and moisture release from the sea (not shown). The specific humidity (Fig. 4f) follows the distribution of SST (Fig. 5a), where the sea is warmer, the humidity is higher. Temperature, specific humidity, and wind speed maintain the same spatial distribution at different vertical levels (950 hPa and 850 hPa) inside the PBL thickness (Fig. S2 in supplementary). A similar distribution of cyclones' composite fields is present in SON (Fig. S3).



**Figure 4: Maps for total precipitation (a), convective precipitation (b), PBL height (c), 10 m wind speed and direction (d), evaporation (e) and 2 m specific humidity (f) from the STD simulation during winter extreme cyclones in common with CPL.**



### 3.2.2 SST analysis

Before examining the differences between the atmospheric fields of CPL and STD, it is crucial to investigate the SST distribution, which is pivotal in controlling evaporation and precipitation (Lebeaupin Brossier et al., 2015) and may underline

the differences between the RCMs. Focusing on the winter season, a clear north-south gradient is visible for SST in the MED-REP-L4 dataset (Fig. 5a) with warmer temperatures in the south-eastern part of the Mediterranean Sea and colder near the French coast and upper Adriatic at the mouth of the Po River. During extreme winter cyclone events, compared to STD, the CLP model is significantly warmer, up to 1.5 °C, over most of the Mediterranean Sea, except for the northern part of the Adriatic Sea and, to a smaller degree, the Eastern Sea where the difference has opposite signs (Fig. 5b). SST differences are

not associated with the occurrence of the cyclones, but rather to the climatological bias of explicitly resolved SST by the coupled model. Indeed, the same bias appears also when comparing the SST climatology in CLP with the MED-REP-L4 dataset, while no significant differences are found between STD and MED-REP-L4 (Fig. 5c and 5d).

All the outcomes on DJF are also valid for the analysis of the SST bias in SON (Fig. S6). However, in this case the SST differences between the models have opposite sign: in fact, CPL is significantly colder than both the observation (Fig. 6c) and

STD (Fig. 6d), except for the strait of Gibraltar, near the south-France coast and the norther part of the Adriatic Sea. Moreover, the magnitude of the SST difference between CLP and STD is substantially reduced compared to DJF over the western Mediterranean, where most of the extreme cyclones are located (Fig. S1 in supplementary).

Further information on the validation of the ocean system of the CPL can be found in Anav et al. (2024) for all the seasons.



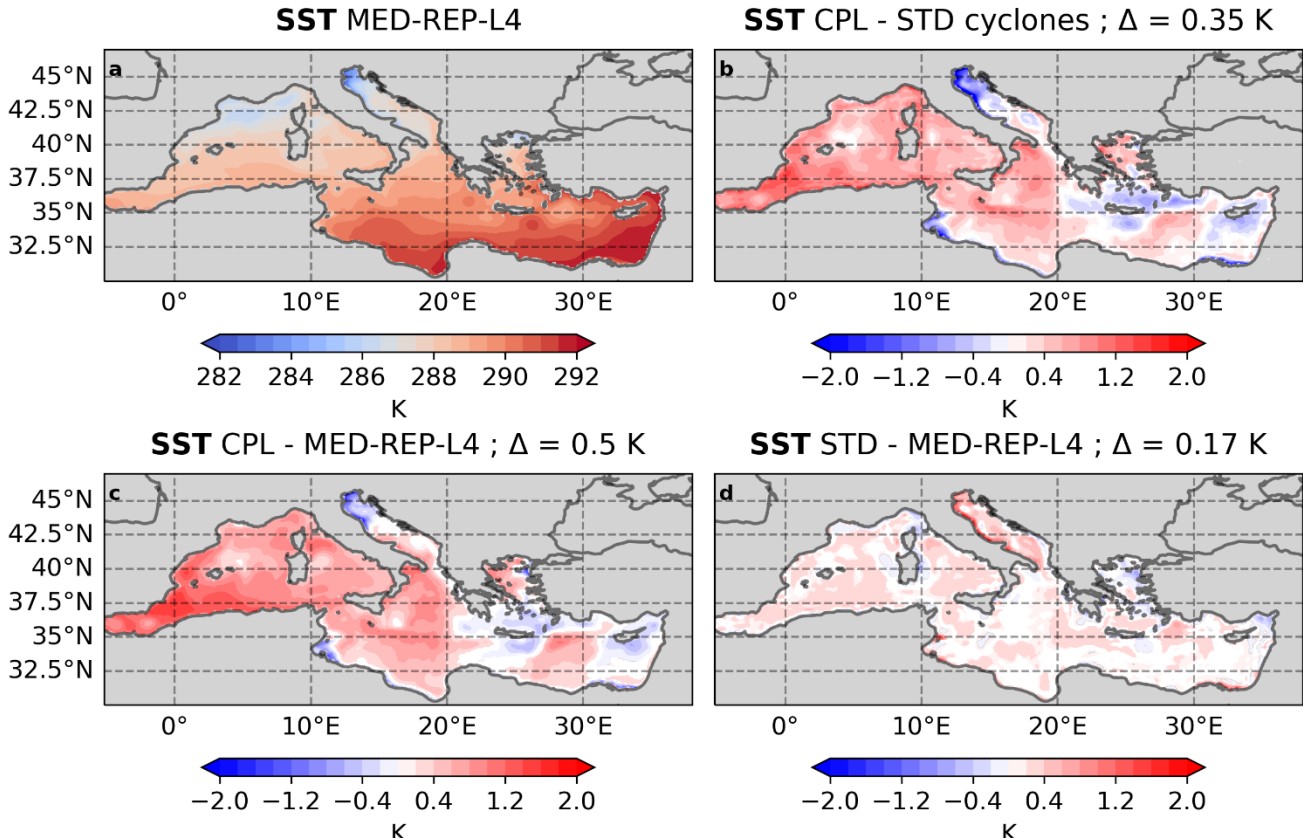

**Figure 5: Maps for the SST from the MED-REP-L4 observational dataset (a), maps of difference in SST between CPL and STD for winter extreme cyclones (b), maps of the winter climatological difference in SST between CPL and MED-REP-L4 (c) and between STD and MED-REP-L4 (d). The white colour indicates no significant differences at 5% confidence level.**



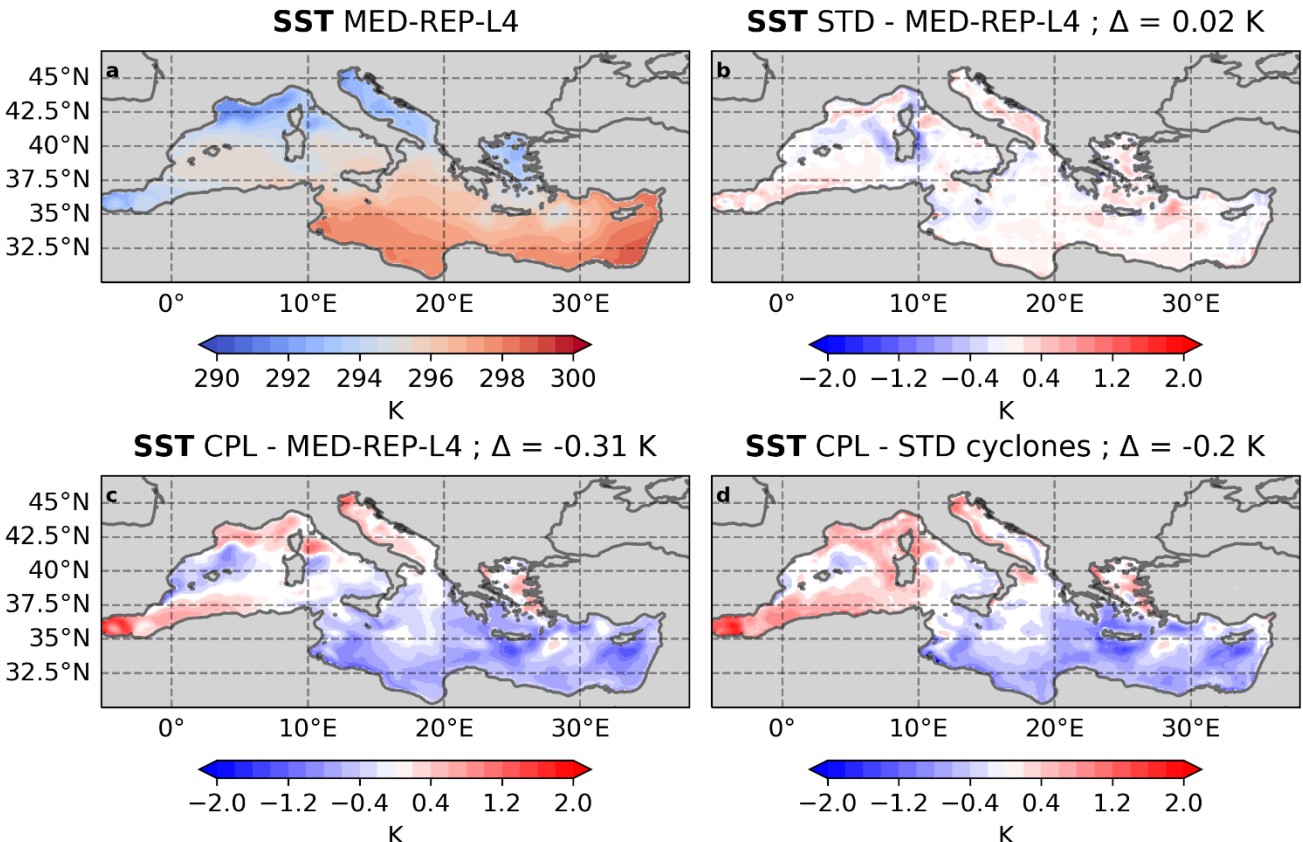

**Figure 6: Same as figure 5, but for SON.**

### 3.2.3 Atmospheric differences between CPL and STD

The impact of ocean-atmosphere coupling and SST distribution on precipitation is indirect and implies several physical processes (PBL turbulent transport, convection, and microphysics), producing a complex rainfall response with positive and negative differences.

The warmer SST of CPL fosters evaporation (Fig. 7a), as also latent and sensible heat fluxes (not shown), which then modifies the atmospheric low levels fields, i.e., the 10 m wind speed (Fig. 7b) and the 2 m specific humidity (Fig. 7c). In the areas of warmer sea, the higher turbulent sensible and latent heat fluxes of the CPL increase the vertical transport, destabilising the PBL and increasing its height (not shown). This is explained by the higher $\Theta$ gradient of the CPL (Fig. 7d), that makes the PBL less stratified and higher, and by the significant positive correlation between SST, evaporation, PBL height and $\Theta$ gradient differences (Fig. 9). This instability enhances the cloud formation (not shown) and thus increases by 15% the simulated convective precipitation (Fig. 7e), as highlighted in Fig. 9 by the high correlation between convective precipitation and both SST and evaporation differences (R=0.7). On the other hand, colder SST, as in the Adriatic and Levantine Sea, induces weaker



evaporation, increases the stability of the PBL (lower Ѳ gradient and lower PBL height) and reduces the 2 m specific humidity, the 10 m wind speed, and the convective precipitation.

While the relation between warmer SST and higher evaporation and convective precipitation is well-known (Flaounas et al.,
2016; Miglietta et al., 2011), the link between warmer SST and higher 10 m wind speed is however not obvious. There is a significant positive correlation between SST and 10 m wind speed differences (R=0.8, Fig. 9). A warmer SST increases the turbulent heat fluxes at the surface (thermally generated turbulence), with warm air rises and cold air sinking due to buoyancy forces, that could help to transfer energy downward to the surface, thus explaining the increase in the 10 m wind speed.

The stronger horizontal winds in CPL lead to a mismatch between areas of high vertical moisture flux and total precipitation
as shown by the lower correlation between SST (evaporation) and total precipitation (R=0.5 (R=0.6), Fig. 9). In fact, during intense wind events associated to the cyclones, the precipitation differences did not come directly from changes in the surface air moisture but from the wind dynamics that are responsible to the changes in the convergence zones of moisture, as discussed in Berthou et al. (2016). It is interesting to note that, considering the winter climatology (not shown), rather than extreme winter cyclones, the seasonal precipitation differences are much better correlated with the climatological differences of SST,
with a Pearson correlation coefficient equal to 0.72 (not shown), similar to the value found by Lebeaupin Brossier et al. (2015) for the Mediterranean Sea (R = 0.74).







**Figure 7: Maps of the differences between CPL and STD during the common extreme winter cyclones for evaporation (a), 10 m wind speed (b), 2 m specific humidity (c), ϴ gradient (d), convective precipitation (e) and total precipitation (f). The white colour indicates no significant differences at 5% confidence level.**

Evaluating the extent to which differences in SST influence atmospheric fields such as temperature, specific humidity, and wind speed at various atmospheric heights is intriguing. In regions with warmer sea, the higher turbulent sensible and latent





heat fluxes from the CPL affect, not only surface atmospheric properties and dynamics, but also modify characteristics throughout the entire PBL. In fact, the CPL remains warmer than STD at 850 hPa, although the temperature differences

between the models decrease with altitude (0.6 K at 950 hPa and 0.1 K at 850 hPa; Fig. 8a, b), and the direct connection with SST dissipate above 950hPa (R=0.6 at 950 hPa and not statistically significant correlation at 850hPa, Fig. 9). The differences in specific humidity between models seem to mimic the temperature differences at both vertical levels, showing a band of higher humidity from the western to the eastern part of the Mediterranean basin (Fig. 8c, d). The correlation between SST and specific humidity remains significant at both vertical levels, although it decreases with altitude (R=0.7 at 950 hPa and R=0.5

at 850 hPa, Fig. 9). Conversely, the differences in wind speed are limited (Fig. 8e, f) and exhibit no correlation with SST at both vertical levels (Fig. 9). Consequently, the SST distribution impact the wind only at the surface, intensifying it where the sea is warmer, due to thermally induced turbulence at the sea surface, as previously discussed.

At 700 hPa (Fig. S4 in supplementary), there are no significant differences in temperature, specific humidity, and wind speed nor correlation with SST differences (not shown) proving that the sea-surface turbulent fluxes do not affect the atmospheric

properties above the PBL.

The same methodology as in winter is also applied to autumn extreme cyclones. In SON, there are only 59 extreme cyclones (i.e. among the 500 most intense cyclones) in common between the CPL and the STD simulation. The analysis in SON agrees with the winter results, i.e. the different SST between CLP and STD impacts the evaporation (Fig. S5a in supplementary), the

stability of the Planet Boundary Layer (THETA gradient, Fig. S5d in supplementary), as well as the specific humidity and temperature at different vertical levels (Fig. S6 in supplementary). However, the differences in SST between CLP and STD (Fig. 6) are substantially reduced compared to DJF over the western Mediterranean, where most cyclones are located in SON (Fig. S1 in supplementary). This leads to not statistically significant differences in 10 m wind speed (Fig. S5b in supplementary) and in convective (Fig. S5e in supplementary) and total precipitation (Fig. S5f in supplementary) (same

bootstrapping method applied for winter) and lower linear correlation between the SST and the atmospheric differences (Fig. S7 in supplementary). This result confirms our conclusion, i.e. the different SST distribution between the models is the dominant factor in shaping both the sea surface fluxes, and the precipitation and wind speed differences associated to the extreme winter cyclones, but also the atmospheric properties at different vertical levels up to the top of the atmospheric boundary layer.

The next session investigates the ability of the CPL model to simulate the impact of the extreme cyclones on the ocean system in both DJF and SON.





**Figure 8:** Maps of the differences between CPL and STD during the common extreme winter cyclones for temperature (a,b), specific humidity (c,d) and wind speed (e,f) at 950 hPa and 850 hPa. The white colour indicates no significant differences at 5% confidence level.

 



**Figure 9: Pearson correlation coefficient (R) between the SST differences (Fig. 5b, CPL – STD) and the differences in the atmospheric fields analysed (Fig. 7 and 7, CPL – STD), during the extreme winter cyclones. The matrix is symmetric, and the empty squares mean not statistical significance of the coefficient (P value equal to zero). Qv stand for specific humidity and T for temperature at different vertical levels.**





### 3.3 Ocean response to extreme cyclones

The previous section showed how the energy at Sea surface affect the atmosphere throughout the PBL during extreme winter cyclones, while here it is evaluated if the coupled model allows redistribution of the turbulent energy generated during these
events, not only within the atmosphere but also into the ocean. This feature would be a key advantage of the coupling system, as it allows for a more comprehensive representation of the thermodynamic processes associated with cyclones across the entire system. More specifically, it enables a coherent modelling of the impacts of such large-scale upper-troposphere instabilities from atmospheric layers down to the ocean layers, within the Mixed Layer depth (MLD), where the turbulent exchange processes take place.


The cyclone's impact on the ocean structures has been investigated both in DJF and SON (spatial distributions of the cyclones in supplementary, Fig. S8), to consider the different states of the ocean in these seasons. In general, in DJF the upper ocean is well mixed, therefore the MLD is deeper than in SON, where the upper sea is still stratified by the seasonal thermocline developed during summer. For both DJF and SON, figure 10 (a, b) shows the SST temporal evolution before and after the
cyclones comparing the CPL model (blue lines) with STD model (green lines), which is forced at the surface by ERA5 reanalysis, and CMEMS MED-Currents reanalysis of the Mediterranean Sea (orange lines). In addition, for CPL and MED-Currents, the figure 10 (c, d) shows how the vertical profiles of the ocean temperature modifies two days before and after with the day of the cyclones. In winter, due to the deep mixed layer, the effect of cyclones on ocean structure is weak, with a very low mean cooling of the temperature at both the surface (Fig. 10a) and at different vertical levels (Fig. 10c). Conversely, in
SON the impact of the cyclones on the ocean structure is stronger with a significant mean cooling that decrease from the surface (Fig. 10b) to the depth of the mixed layer (Fig. 10d). In autumn, the shallower mixed layer and the ocean stratification favour the upwelling processes caused by the strong winds during cyclones that enhance evaporation and surface heat releases. This results in a cooling of the surface water, which becomes denser and sinks (density increasing in Fig. S9 in supplementary), increasing the MLD and the turbulent mixing processes.


It is interesting to note that, despite the SST bias (Fig. 5 for DJF and Fig. 6 for SON), the CPL model is able to accurately simulate the impact of extreme cyclones on the ocean temperature evolution at the surface (Fig. 10a and b) and throughout the MLD (Fig. 10c and d), being very close to the MED-Currents reanalysis in all cases. Thus, in both seasons, the CPL model reproduces the cooling effect of the cyclones on the SST better than STD, although the SST distribution of the latter comes
directly from the ERA5 reanalysis dataset, which is closer to both the climatological SST of MED-REP-L4 dataset (DJF, Fig. 5b and SON, Fig. 6b) and to MED-Currents reanalysis (Fig S10, DJF (b) and SON (d), in supplementary).
This analysis proves the ability of the high-resolution coupling system to coherently simulate both the atmospheric and ocean processes associated to the Mediterranean cyclones, which is a crucial aspect for climate change studies when the SST cannot be corrected with observations.



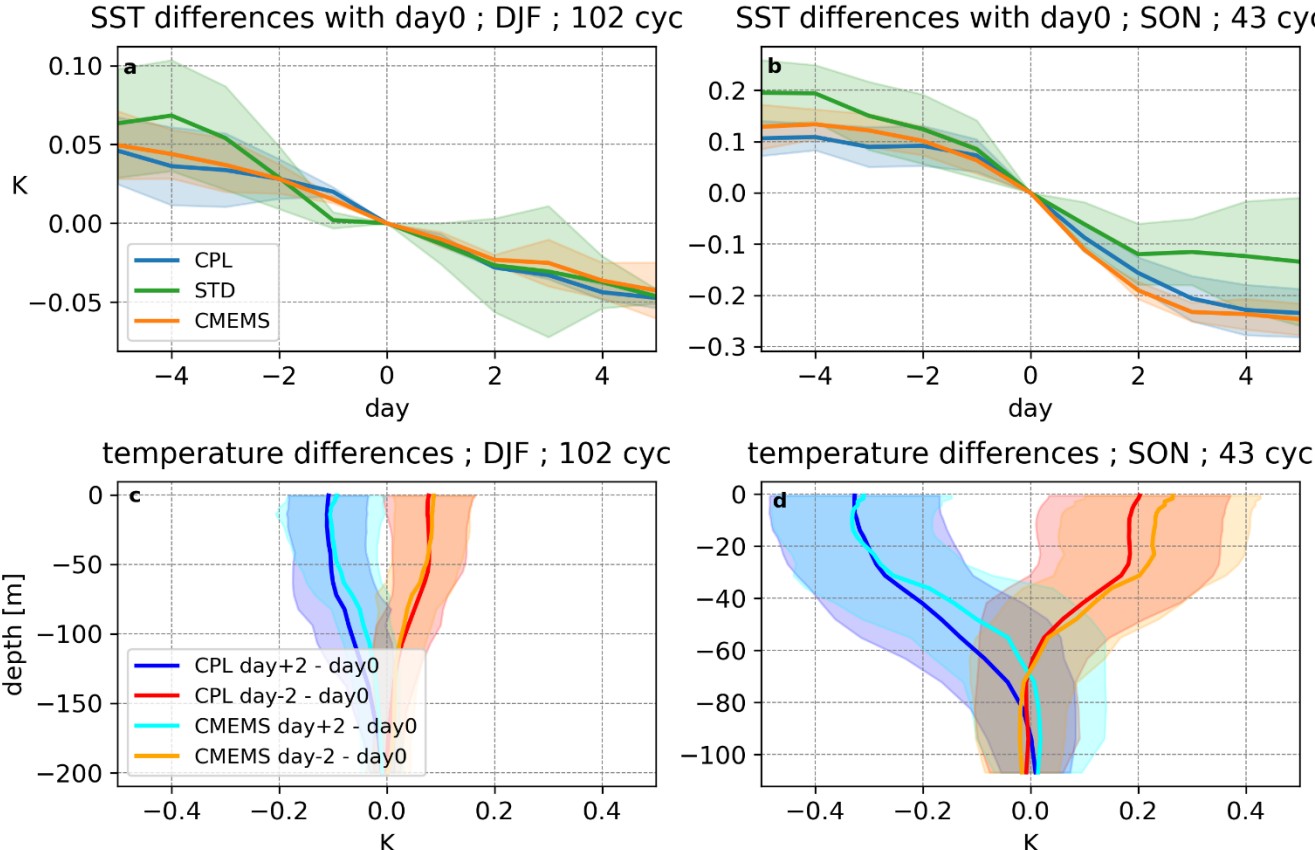

**Figure 10: SST evolution compared with the SST on the day of the cyclone from five days before to five days after the event for CPL (blue line), STD (green line) and CMEMS MED-Currents reanalysis (orange line), averaged over the same cyclones in DJF (a) and in SON (b). The vertical profiles of the ocean temperature computed as difference between 2 days before and the day of the cyclones (similarly for 2 days after the event) for CPL (blue and red lines) and CMEMS MED-Currents (light blue and orange lines), averaged over the same cyclones in DJF (c) and in SON (d). The colour bands represent the confidence interval between +- 1 standard deviation of the mean of the temperature differences.**



## 4 Discussion and conclusion

This study employed two high-resolution RCM simulations, one atmosphere-ocean coupled (CPL) and one atmosphere stand-alone (STD), to investigate the climatology of extreme Mediterranean cyclones and the influence of the explicitly resolved

SST distribution and sea surface fluxes in the CPL model on the atmospheric and ocean fields simultaneously during the extreme cyclones over the period 1982-2014. The results indicate that, while extreme cyclones significantly influence the Mediterranean climate, the coupling between the atmosphere and ocean exerts a limited influence on their climatology and statistics (i.e. lifetime, speed, and intensity). This aligns with previous studies demonstrating that the coupling system has a limited effect on the climatology of the Mediterranean cyclones (Flaounas et al., 2018b), because they are predominantly

driven by large-scale upper-tropospheric forcings (Chaboureau et al., 2012; Fita et al., 2006; Flaounas et al., 2015, 2021; Neu

et al., 2013; Raveh-Rubin and Flaounas, 2017). However, when comparing CPL and STD, it becomes evident that the different SST distribution between the models is the dominant factor in shaping both the sea surface fluxes and the precipitation and wind speed differences during the passage of extreme winter cyclones. More specifically, the warmer SST in CPL fosters higher evaporation and surface heat fluxes, leading to modifications in atmospheric properties, such as temperature and specific

humidity, up to the top of the boundary layer. The higher turbulent fluxes increase both the 10 m wind speed, due to the higher energy at the surface, but also the convective precipitation, destabilizing the boundary layer and providing more energy to sustain convection.

In the CPL, the fluxes of heat and moisture and the wind speed, increased during the extreme cyclone events, affect not only the atmosphere but also the ocean properties. The strong winds across the ocean enhance the surface fluxes and favour the

upwelling of the colder waters, increasing the turbulent mixing processes and resulting in a cooling effect on the ocean temperature throughout the entire mixed layer. Despite the climatological bias of the SST, the CPL model better represents the cooling effect of the cyclones on the SST compared to STD and, in addition, accurately simulate the ocean response to these events. In fact, the temporal variation of the ocean temperature from the surface down to the mixed layer depth, during the passage of the cyclones, simulated in the CPL model is very close to that of the CMEMS MED-Currents reanalysis.


This research represents a step forward on how the energy, generated during extreme cyclones, redistributes from the sea surface up to the atmospheric boundary layer height and down to the ocean mixed layer depth. The coupled model allows to represent the cyclones' impacts on convective precipitation and surface wind speed as well as on ocean temperature and density. This enhances our confidence on the ability of the coupled model to coherently represent the entire atmosphere-ocean

system under extreme events associated to cyclones. This finding is of crucial importance in the climate change context since atmosphere-ocean coupled RCMs give the possibility to remove the uncertainty deriving from coarse resolution SST coming from the global models.

**Author contribution**

M.C. performed the analysis on the RCMs climate data and wrote the manuscript with inputs from all the authors. G.S.

developed the CPL model. A.A. performed the simulations. E.F. developed the storm track algorithm.

**Competing interests**

The authors declare that they have no conflict of interest.



**Funding**

This study was carried out within: ICSC Italian Research Center on High-Performance Computing, Big Data and Quantum

Computing and received funding from the European Union Next-GenerationEU (National Recovery and Resilience Plan-NRRP, Mission 4, Component 2, Investment 1.4-D.D: 3138 16/12/2021, CN00000013). RETURN Extended Partnership and received funding from the European Union Next-GenerationEU (National Recovery and Resilience Plan NRRP, Mission 4, Component 2, Investment 1.3-D.D. 1243 2/8/2022, PE0000005). CoCliCo (Coastal Climate Core Service) research project which received funding from the European Union's Horizon 2020 Research and Innovation Programme under Grant agreement

No. 101003598. CAREHeat (deteCtion and threAts of maRinE Heat waves) project, funded by the European Space Agency (ESA, grant agreement no. 4000137121/21/I-DT).

**Acknowledgements**

This paper and related research have been conducted during and with the support of the Italian inter-university PhD course in Sustainable Development and Climate change (link:www.phd-sdc.it) and developed within the framework of the project

"Dipartimento di Eccellenza 2023-2027", funded by the Italian Ministry of Education, University and Research at IUSS Pavia. We acknowledge the World Climate Research Programme, which, through its Working Group on Coupled Modelling, coordinated and promoted CMIP6. Within this we thank the CMIP6 endorsement of the High-Resolution Model Intercomparison Project (HighResMIP) and Martin Schupfner for providing additional data from the MPI-ESM. The computing resources and the related technical support used for this work have been provided by CRESCO/ENEA-GRID High

Performance Computing infrastructure and its staff.
This article is based upon work from COST Action CA19109 "MedCyclones", supported by COST - European Cooperation in Science and Technology (http://www.cost.eu).
We acknowledge the Copernicus Marine Service and the CNR – ISMAR for the data provided for our analysis, MED-REP-L4 and CMEMS MED-Currents, respectively.

We would like to thank you Antonio Segalini from Uppsala University and Marina Tonani from the Mercator Ocean International non-profit organisation for their valuable insights and discussions on the results.

**Data availability**

Enquiries about data availability should be directed to the authors.





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
