# Peer review of "Extreme Mediterranean cyclones and associated variables in an atmosphere-only vs an ocean-coupled regional model"

_EGUsphere, 2024_

## Referee Comment (RC3)

Extreme Mediterranean cyclones and associated variables in an atmosphere-only vs an ocean-coupled regional model
WCD-2024-2829

This paper aims to quantify the impact of air-sea interactions during extreme cyclone events on the structure of the atmospheric and oceanic boundary layers. This is addressed by performing two 33-year simulations using coupled (CPL) and atmosphere only (STD) models. First the cyclone climatologies from the CPL and STD simulations are compared to an atmospheric reanalysis dataset (ERA5). Then the climatological SST fields during extreme cyclones from the CPL and STD are compared to satellite-based SST dataset (MED-REP-L4). Next, the CPL and STD atmospheric fields are compared and finally the evolution of the ocean structure in the CPL simulation is compared to and ocean reanalysis dataset (CMES). I found the paper a bit confusing to read and the grammar is incorrect in many places. The motivation for the analysis and importance of the study needs more emphasis before the paper can be considered to be suitable for publication.

General comment
1. There are fairly large differences between the CPL/STD cyclone climatologies and ERA5, shown in figs 2 & 3, which are dismissed as small in the paper. Do these differences result in differences in the cyclone-climatology atmospheric fields (precipitation, pbl height, 10m wind speed, evaporation and 2m specific humidity)? Figure 4 does not compare with ERA5 fields so it's not possible to determine the answer to this question.
2. There are large differences between CPL and satellite-based SST cyclone climatology fields (0.5K) shown in figs 5 and 6. Unsurprisingly, these differences go on to dominate the spatial maps shown in figures 7 and 8, as demonstrated by the high correlations in figure 9. I was missed the importance of this result. The authors have demonstrated that differences in SSTs leads to large differences in the atmospheric fields, but what is the link to the extreme cyclones. Are the SST differences larger during cyclone events than non-cyclone events, and thus accurate prediction of extreme cyclones is important? If so, do the non-cyclone climatologies also need to be included to demonstrate this?
3. Figure 10 is the most interesting result because it removes the removes the bias in SST and thus allows a comparison of the effect of coupling. What causes the difference between CMEMS and STD SST evolution (using daily ERA5 SST) prior to the cyclone event?

Specific comments
1. Line 12, 13: ENEA-REG, Med-CORDEX, ERA5, WRF, MITgcm acronyms need to be defined. Is it important for the general reader to know the names of these datasets and models? If not, please consider writing the abstract using more general language and leave the detailed acronyms to the main body of the paper.
2. Line 24: What do the authors mean by the 'effectiveness' of the coupled model?
3. Line 54: Which side of the Alps is the 'leeward side'? Surely, this depends on the wind direction?
4. Line 80: What do the authors mean by 'proper' air-sea coupling effects?
5. Line 107: Why is the role of SST and air-sea fluxes on extreme events expected to be stronger in the Autumn season?
6. Line 163: 500 cyclones represent almost 20% of the cyclone distribution. This does not seem particularly extreme.
7. Line 175: I found the terminology $\theta$-gradient ambiguous. Why not use static stability or potential temperature lapse rate which are more standard terms for such a metric?
8. Line 216: Why is a radius of $1.5^\circ$ around the cyclone centre chosen? This seems to suggest that the enhanced surface fluxes occur very close to the cyclone centre and are axisymmetric. Is this supported by any analysis?
9. Figure 2: This figure shows that there is some seasonal dependence on the performance of the CPL and STD model simulations when compared to ERA5 for predicting the number of cyclones. Is there also seasonal dependency in the other statistics (intensity, lifetime and speed)?

10. Figure 3: The maxima in frequency of cyclones in CPL model, over land, does not correspond to maxima in the STD simulation. Furthermore, the frequency over the ocean reduces in the CPL simulation. Therefore, I do not think the evidence supports the statement on line 230 that 'the spatial distribution of extreme cyclones is similarly reproduced by the models compared to ERA5' or on line 241 that 'differences between STD and CPL are limited and non-significant'. Perhaps difference plots would demonstrate the similarity or differences in the simulations more clearly?
11. Line 264: How do the cyclones 'turn into precipitation when they reach the coast'? Do you mean that at the coast, orographic ascent causes water vapour to be converted into water droplets, which then grow into precipitation droplets?
12. Line 267: Can the authors expand on their statement that the transition between sea and land fosters the convection processes? Are you referring to convergence at the coast?
13. Figure 4: How do these composites differ from the full winter climatology? Perhaps anomaly fields could be shown?
14. Fig 6: The order of the figures is different to that in figure 5 which confused me for a while.
15. Fig 7: The order of the figures is different to those in fig 4. Could they be reordered to be consistent?
16. Figure 8: Since these fields are similar to those shown in previous figures, I do not think they add much to the analysis.
17. Line 304: Reference is made to latent and sensible heat fluxes. Could these fields be shown instead of evaporation, wind speed, specific humidity and theta gradient? They are directly responsible for increasing the vertical exchange of heat and moisture between the Mediterranean sea and overlying atmosphere so would be more relevant.
18. Figure 10: It appears that the MLD deepens more in the CPL model than in the CMEMS, why is this?
19. Figure 10: It is interesting that the MLD is twice as deep in winter than in autumn. Is this why the change in SST is so much smaller in winter?
20. Line 417: Is there evidence to support the statement that extreme cyclone significantly influences the Mediterranean climate? By climate, do the authors mean the long-term average conditions?

Typographical errors
1. Line 20: 'Planet' should be 'planetary'.
2. Line 20: 'mixing of the turbulent processes' should be 'mixing by the turbulent processes'.
3. Line 98: 'insights on how' should be 'insights into how'.
4. Line 117: 'than STD' should be 'as STD'.
5. Line 174: 'planet boundary layer' should be 'planetary boundary layer'.
6. Line 179: What is the 'e' after STD? Is this a typographical error?
7. Line 226: 'upscaled at ERA5 resolution' should be 'upscaled to ERA5 resolution'
8. Line 290: 'norther' should be 'northern'.
9. Line 350: 'THETA' is represented as $\theta$ elsewhere.
10. Line 53: 'Not statistically significant differences' should be 'statistically insignificant differences'.

---

## Author Comment (AC1)

**Reviewer #1:**

**This manuscript uses a cyclone tracking algorithm to track the most intense Mediterranean cyclones in atmosphere-only and coupled atmosphere-ocean simulations. It is shown that both STD and CPL simulations represent the climatology of storms in the Mediterranean with no notable advantage to CPL. Also, it is shown that CPL has an SST bias relative to STD, which affects various fields in the PBL. CPL can be used to understand dynamical mechanisms in the ocean mixed layer in the presence of atmospheric cyclones.**

**Overall, the work performed for this study is impressive – a combination of coupled model simulations with cyclone tracking algorithms and the subsequent analysis. The presentation of the results and the discussions are interesting and well-structured. Yet, I struggle to see the innovative part of this paper. Instead, I see a nice comparison between two simulations – coupled and uncoupled. Ultimately, the primary distinction between simulations is the SST bias in CPL, which imprints on various fields in the PBL. The relevancy of the coupling vs. non-coupling is only demonstrated in Figure 10, Panels a and b. There, the advantage of coupling is clear.**

**I think it would be helpful to use more careful language that does not attribute the difference of various fields to the coupling (or the explicitly resolved SST). These suggested changes perhaps mean that the conclusions are more relaxed. Still, considering the large effort made by the authors, after addressing the above critics and the more specific comments below, I would recommend the paper for publication.**

We thank the Reviewer for dedicating time to review our manuscript and for the detailed observations that have raised the quality of the paper.

The valuable feedback has helped us improve the clarity of the work and better emphasise its results and novelty. Specifically, we have significantly revised the Results section to provide a more detailed explanation of the physical mechanisms behind the differences in the atmospheric processes between CPL and STD during extreme Mediterranean cyclones and to address the specific questions raised by the reviewer.

In addition, we have revised the conclusion to emphasize the novelty of the study. Specifically, we modified lines 432-434, as follow:

"*This study investigates for the first time (to the best of our knowledge) how extreme Mediterranean cyclones affects simultaneously the atmosphere and the ocean at different vertical levels, comparing two high-resolution RCM simulations, one atmosphere-ocean coupled (CPL) and one atmosphere stand-alone (STD), over the period 1982-2014.*",

and lines 454-456, as follow:

"*This research highlights the ability of the coupled model to coherently simulate the entire atmosphere-ocean system, thus providing new insights into how sea surface energy is redistributed between the atmospheric boundary layer and the ocean mixed layer, and how this impacts the precipitation and the wind speed during extreme cyclone events.*".

Please note that figures 4, 7 and 9 have been revised following the reviewers' comments. The figures now show latent and sensible heat fluxes, 10 m wind speed, potential temperature lapse rate, convective precipitation and total (large scale + convective) precipitation.

Please also note that the text-line references mentioned in our responses correspond to the revised manuscript.

Below, the Reviewer will find our detailed, point-by-point, answers.

**Specific comments:**

**Lines 103-104: The authors try to answer the question, "To which extent in the vertical column, and through which physical mechanisms, the explicitly resolved SST distribution and sea surface fluxes impact the precipitation, and the wind speed during extreme cyclones?", but the designed simulations can't really separate the effect of explicitly resolved SST when the SST in the Western Mediterranean is about 1.5 degrees warmer. In that case, I think the only thing that can be done is to downgrade the question to something that fits the analysis in the paper.**

We thank the Reviewer for highlighting this. We have revised the second research question to better align with the scope of the analysis. The updated question is (L103-105):

"*How do differences in SST distribution shape atmospheric processes within the planetary boundary layer (PBL) during extreme cyclone events, and how do these mechanisms, in turn, impact the cyclone-related precipitation and wind speed?*"

**Line 167-168: "Two cyclones are considered the same event if their minimum of SLP is within a 500 km distance and within a time range of 12 hours." – this stage removes from the analysis all cases in which CPL is different from STD (~30%). This difference by itself sounds very large, suggesting that many cyclones are represented very differently in the CPL simulation. The authors do not compare their tracks with ERA5-based tracks (e.g., by calculating the RMSE of the distance between observed and simulated cyclone location at maximum intensity), and it is hard to tell which simulation is better. Therefore, it may lead to an unverified conclusion that CPL represents the cyclones well (although it has a large SST bias) and that CPL does not have an added value.**

We thank the Reviewer for the valuable feedback. First, we would like to point out that there was a typo error in the manuscript: the same extreme cyclones between CPL and STD are 312 and not 341. This corresponds to approximately 62 % cyclones of the 500 most intense cyclones.

We agree with the Reviewer's observation that the extreme cyclones in common between STD and CPL can appear substantially different. However, similar differences were previously found also in Flaounas et al. 2018a (around 60 % of common cyclones). In fact, differences in SST fields influence the distribution of the SLP, even if it is forced by the same large-scale atmospheric condition (ERA5 in this case). This does not imply that the cyclones themselves are fundamentally different but rather that a different SST may affect the timing and location of the cyclones SLP minima. To further test this statement, we changed the 500 km distance and the 12-hour time window criterion, that qualifies two cyclones as being "the same" in the two simulations, with 1000 km (i.e. the maximum area of influence of Mediterranean cyclones, Flaounas et al. 2016) and 48-hour. In this case, we found 458 common events (92%) between CPL and STD. For these 458 cases, we repeated the same analysis (shown in Figure A1 and attached below). The results are very similar to those obtained for the original 312 common cases (Fig. 7), indicating that our main results are not sensitive to the chosen constraints or to the number of selected "common" cyclones.

Regarding the track comparison between RCMs and ERA5, our objective is not to determine whether the CPL or STD tracks better match ERA5-based tracks but to better understand their development mechanisms regards to air-sea interactions. Nevertheless, we calculated the RMSE and spatial correlation (R, Pearson correlation coefficient) for the locations of the minima of the 500 most intense cyclones in CPL and STD against ERA5. The metrics are similar for both models, with CPL showing slightly lower RMSE (2.16 vs. 2.17 in STD) and slightly higher R (0.73 vs. 0.72 in STD). This is attributed to CPL's higher frequency of cyclones in the Tyrrhenian Sea, which aligns more closely with the ERA5 distribution (see Fig. 3b vs. 3c).

We have clarified in section 2.2.2 (model comparison) the methodology applied to identify common cyclone events, as follow (L176-184):

*"To compare CPL with STD in terms of sub-daily fields associated to the cyclones, the same events between the two simulations are selected. Two cyclones are considered the same event if their minimum of SLP is within a 500 km distance and within a time range of 12 hours. With these criteria, a total of 312 cyclones from the 500 most intense (around 62 %) are found in common between CPL and STD, of which 129 occurring in winter (DJF), 110 in spring (MAM), 17 in summer (JJA) and 56 in autumn (SON). This well align with results from Flaounas et al. (2018a), who also found that approximately 60% of the 500 most intense cyclone tracks were consistent between the coupled and standalone RCMs, using similar identification criteria. Extending the distance criterion to 1000 km (i.e. the maximum area of influence of Mediterranean cyclones, Flaounas et al. 2016) and the time window to 48-hour, the percentage of detected cyclones in common between STD and CPL increases to 92%, but the outcomes of this study do not change (not shown)."*

[Figure]

**Figure A1: Same as figure 7 but for the 458 common cyclones between CPL and STD.**

**Line 172-173: does this mean that out of 341, 199 cyclones occur in DJF and SON? Please clarify this.**

We have revised lines 172-173 to clarify the seasonal distribution of the same cyclone events as follows (L178-180):

"*With these criteria, a total of 312 cyclones from the 500 most intense (around 62 %) is found in common between CPL and STD, of which 129 occurring in winter (DJF), 110 in spring (MAM), 17 in summer (JJA) and 56 in autumn (SON).*"

**Line 176: can you explain or provide a reference for why this field at the specific level was chosen?**

In the revised manuscript we have included in the analysis only the temperature and specific humidity fields at both 950 hPa and 850 hPa levels (Fig. 8). These two levels were selected to look at two different atmospheric layers within the PBL, allowing us to assess the impact of the SST differences not only near the surface but throughout the entire PBL. The 950 hPa level (approximately 500 m altitude) is widely used to analyse atmospheric low-level properties (e.g. Raveh-Rubin and Wernli, 2016), while 850 hPa level (approximately 1500 m altitude) is close to the top of the PBL, providing insight into upper part conditions of the PBL (e.g. Fosser et al. 2015).

*Fosser, G., S. Khodayar, and P. Berg, 2015: Benefit of convection permitting climate model simulations in the representation of convective precipitation. Clim. Dyn., 44, 45–60, doi:10.1007/s00382-014-2242-1.*

**Figure 3: It is not clear how this figure was made. Is the percentage calculated from all days in the specific grid cell or is it a percentage from all cyclones in the region? Could you please clarify this? Also, while the spatial variability is well represented in the models, there is still a very large difference in the percentage relative to observations. The authors should discuss this, at least by providing some information about the source of this large difference.**

We thank the Reviewer for pointing out the need for clarification. We have revised both figure 3 and section 3.1 to better clarify the results on the comparison between RCMs and ERA5 500 most intense cyclones. We have also provided references on the methodology applied and during the discussion of the results.

Figure 3 is now computed as follow (L247-251):

"*Figure 3 instead shows the maps of cyclone centre densities (CCD; Neu et al. 2013, Flaounas et al. 2018a) for ERA5 and the differences in CDD between ERA5 and RCMs. The CCD is defined as the absolute number of occurrences of the 500 most intense cyclone centres. To highlight the cyclones' area of influence, each centre is represented by a circular area with radius of 1.5 degrees around the tracked minimum SLP point.*"

[Figure]

**Figure 3: Number of occurrences of cyclone centre densities (CCD) for the 500 most intense cyclones in ERA5 (a), along with CCD differences between STD and ERA5 (b), CPL and ERA5 (c), and CPL and STD (d). To highlight the cyclones' area of influence, each centre is represented by a circular area with radius of 1.5 degrees around the tracked minimum SLP point.**

The discussion about the differences between RCMs and ERA5 has been modified as follow (L251-271):

"*Compared to ERA5, both RCMs tend to capture the main regions of frequent cyclogenesis (over the gulf of Genoa, over the Adriatic and Aegean Seas and the marine areas close to Cyprus). This can be expected since the most intense Mediterranean cyclones are formed due to large scale forcing, i.e. the intrusion of upper tropospheric systems as a result of Rossby wave breaking over the Atlantic Ocean (Flaounas et al., 2022). This upper tropospheric forcing is identically introduced to the two simulations through the boundary conditions. However, cyclones seasonality and location also depend on diabatic forcing due to convection within the cyclone systems, as well as on the basin's orography. Both RCMs show a higher occurrence of cyclones in summer and spring (Fig. 2d and Fig. S1 in supplementary) and compared to ERA5, they tend to underestimate the CCD over the Mediterranean Sea while overestimating it over land and over the Aegean and Levantine Sea (Fig. 3b and c). Differences between the two RCMs and ERA5 arise primarily from the different resolution, dynamics and physical parameterisation. These factors influence how the models reproduce key processes, such as. the impact of orography on cyclone dynamics and the role of convection in deepening the cyclones, resulting in local deeper minima of SLP over Mediterranean areas with complex land-sea distribution. Therefore, differences from ERA5 should not be taken purely as a weakness of RCMs, but rather as a result of differences when reproducing atmospheric processes. Indeed, the magnitude of these differences is comparable to the one found in previous studies (Flaounas et al., 2018a; Reale et al., 2022) and thus RCMs should be expected to deviate from reanalysis. In contrast, changes in the SST distribution have a minor impact on the dynamics of the cyclones, leading to small differences between STD and CPL, primarily in the location of cyclone minima over the sea (Fig. 3d). In conclusion, cyclone systems arise from a combination of large-scale processes (external to the cyclone) and small-scale processes (internal to the cyclone). In this context, atmosphere-ocean coupling is expected to have a stronger influence on the physical processes within the cyclone systems, and a rather weaker effect on their formation, distribution, and track characteristics.*"

**Line 307: "This is explained by the higher Θ gradient of the CPL (Fig. 7d), that makes the PBL less stratified and higher" – what is exactly explained by the higher Θ gradient, and does this gradient is the reason why PBL is less stratified and higher? I would say that this is because of the higher SST, as mentioned in the previous sentence. This reasoning is not clear to me. I would say that, in general, the PBL should be well mixed, and differences between STD and CPL should be pretty small in terms of the temperature gradients inside the PBL. I would attribute the difference only to the SST difference.**

We agree with the Reviewer's comment. It is indeed the warmer SST in the CPL model that causes the PBL to be higher and less stratified (stable). We demonstrated this by showing the differences in the potential temperature lapse rate between CPL and STD (Fig. 7d). To note that we have updated the terminology from "Θ gradient" to "potential temperature lapse rate", according to Reviewer 3's comment.

To better clarify the atmospheric processes occurring during the extreme cyclones we have revised the manuscript (L348-352) as follow:

"*In regions with warmer sea, the higher sensible and latent heat fluxes in the CPL model affect, not only surface atmospheric properties, but also modify atmospheric characteristics throughout the entire PBL. In fact, the CPL remains warmer and moister at both 950 hPa and 850 hPa (Fig. 8), and the vertical transport processes of energy are intensified, destabilising the PBL. This is proved by the lower potential temperature lapse rate in the PBL of the CPL model (Fig. 7d), indicating reduced stratification and stability*."

**Figure 10: It is unclear which region is considered when calculating the SST difference. Is it one grid cell where maximum cyclone intensity occurred, or is it a regional average?**

We thank the Reviewer for highlighting the need for further clarification. The method used to compute Figure 10 was originally explained only in the methodology section. To ensure clarity in Section 3.5 ("Ocean response to extreme cyclones") as well, we have added the following sentence on lines 398-400:

"*For each cyclone the ocean temperature is averaged over a circular area with 1.5° radius, around the minimum SLP tracking point and then averaged over the cyclones considered*."

In addition, we have provided this information in the caption of Figure 10 as follow:

"*SST evolution compared with the SST on the day of the cyclone from five days before to five days after the event for CPL (blue line), STD (green line) and CMEMS MED-Currents reanalysis (orange line), averaged over the same cyclones in DJF (a) and in SON (b). The vertical profiles of the ocean temperature computed as difference between 2 days before and the day of the cyclones (similarly for 2 days after the event) for CPL (blue and red lines) and CMEMS MED-Currents (light blue and orange lines), averaged over the same cyclones in DJF (c) and in SON (d). In each figures the temperature values represent the average over a circular area with 1.5° radius, around the minimum SLP tracking point, and over the cyclones considered. The colour bands represent the confidence interval between +- 1 standard deviation of the mean of the temperature differences*."

**Figures: can you explain what the deltas at the top of the panels mean? is it a simple domain average?**

The deltas represent the domain-averaged differences only where the values are statistically significant. In the revised manuscript, deltas are shown exclusively for the SST maps (Fig. 5 and 6), and we have updated the figure captions to include an explanation of how the deltas are computed.

---

## Author Comment (AC2)

**Reviewer #2:**

**The present study investigates the Mediterranean cyclones using regional climate models (RCMs) as well as the ERA5 reanalysis. They assess the reproducibility of intense Mediterranean cyclones in their atmosphere-ocean coupled RCM simulation and compare the simulation with an atmosphere stand-alone simulation to examine the effect of air-sea coupling in RCMs. They also investigate the impacts of intense Mediterranean cyclones on the ocean using the coupled simulation in comparison to the ocean observations.**

**Overall, this study potentially provides materials with implications to improve our understanding of the importance of air-sea interactions for Mediterranean cyclones, which hold significant socio-economic relevance. The influence of air-sea coupling assessed through their RCM simulations is clear and interesting**

**I am afraid, however, that a major revision is needed before this study can be published, for the specific reasons shown below. I am particularly concerned about the interpretation of the results.**

We thank the Reviewer for dedicating time to review our manuscript and for the detailed observations that have raised the quality of the paper.

The valuable feedback has helped us improve the clarity of the work and better emphasise its results and novelty. Specifically, we have significantly revised the Results section to provide a more detailed explanation of the physical mechanisms behind the differences in the atmospheric processes between CPL and STD during extreme Mediterranean cyclones and to address the specific questions raised by the reviewer.

In addition, we have revised the conclusion to emphasize the novelty of the study. Specifically, we modified lines 432-434, as follow:

"*This study investigates for the first time (to the best of our knowledge) how extreme Mediterranean cyclones affects simultaneously the atmosphere and the ocean at different vertical levels, comparing two high-resolution RCM simulations, one atmosphere-ocean coupled (CPL) and one atmosphere stand-alone (STD), over the period 1982-2014.*",

and lines 454-456, as follow:

"*This research highlights the ability of the coupled model to coherently simulate the entire atmosphere-ocean system, thus providing new insights into how sea surface energy is redistributed between the atmospheric boundary layer and the ocean mixed layer, and how this impacts the precipitation and the wind speed during extreme cyclone events.*".

Please note that figures 4, 7 and 9 have been revised following the reviewers' comments. The figures now show latent and sensible heat fluxes, 10 m wind speed, potential temperature lapse rate, convective precipitation and total (large scale + convective) precipitation.

Please also note that the text-line references mentioned in our responses correspond to the revised manuscript.

Below, the Reviewer will find our detailed, point-by-point, answers.

**Specific comments:**

**1 Turbulent heat fluxes, precipitation (and associated diabatic heating), and low-level temperature distribution are important factors for extratropical cyclone development. This study argues that the different SST distribution between the models, which is due to the air-sea coupling, is the dominant factor in shaping anomalies of those variables. Yet, this study also concludes that the coupling between the atmosphere and ocean exerts a limited influence on their statistics such as frequency, lifetime,**

**speed, and intensity. To me, the two conclusions appear inconsistent. The authors should provide a detailed discussion on why these results emerged, addressing both the model's role and potential underlying mechanisms.**

We thank the Reviewer for point out this issue. Here, we clarify the results of our study, addressing the key mechanisms associated to the dynamics of the cyclones.

In **Section 3.1** (*Climatology of Extreme Mediterranean Cyclones*), we analysed the differences between the two models (STD and CPL) and ERA5 in representing the main characteristics of the 500 most intense cyclones. Our findings showed that both regional climate models (RCMs) effectively represent key statistical features of cyclones, including their seasonal cycle, and cyclone track characteristics. However, the models differ from ERA5 in the simulation of cyclones location, mainly due to differences in resolution and model dynamics and physics. These differences affect how the RCM models reproduce e.g. the impact of orography on cyclone dynamics or the role of convection in deepening the cyclones, resulting in local deeper minima of SLP over Mediterranean areas with complex land-sea distribution. In contrast, differences between STD and CPL are minor, leading to the conclusion that atmosphere-ocean coupling exerts only a limited influence on the climatological and statistical properties of extreme Mediterranean cyclones.

This should be a rather "expected" result. Indeed, the most intense Mediterranean cyclones are formed due to large scale forcing, i.e. the intrusion of upper tropospheric systems as a result of Rossby wave breaking over the Atlantic Ocean (Flaounas et al., 2022). This upper tropospheric forcing is identically introduced to the two simulations through the boundary conditions. Furthermore, their characteristic length scale allows their realistic reproduction even at relatively coarse resolutions as the ones in ERA5. The development of cyclones though might depend on both the large scale forcing and diabatic forcing due to convection within the cyclone systems. The latter is strongly dependent on the parametrisation, resolution and -therefore- on the underlying SST. In such climate-scales numerical experiments, one should expect thus that cyclones formation should be rather "similar" in the two simulations, but cyclones development might change at the extent that an extreme cyclone is more diabatically driven than developed due to baroclinic instability (i.e. due to the upper level forcing which should be less sensitive to SST). As a conclusion, if we regard a cyclone system as the outcome of large-scale processes (external to the cyclone system) and small-scale processes (internal to the cyclone system), then we should expect atmosphere-ocean coupling to have a stronger effect on the physical processes of the cyclone systems, and a rather weaker effect on their formation distribution and track characteristics. We have modified Fig. 3 and included this discussion in the revised manuscript (section 3.1, "Climatology of extreme Mediterranean cyclones") as follow (L247-271):

"*Figure 3 instead shows the maps of cyclone centre densities (CCD; Neu et al. 2013, Flaounas et al. 2018a) for ERA5 and the differences in CCD between ERA5 and RCMs. The CCD is defined as the absolute number of occurrences of the 500 most intense cyclone centres. To highlight the cyclones' area of influence, each centre is represented by a circular area with radius of 1.5 degrees around the tracked minimum SLP point. Compared to ERA5, both RCMs tend to capture the main regions of frequent cyclogenesis (over the gulf of Genoa, over the Adriatic and Aegean Seas and the marine areas close to Cyprus). This can be expected since the most intense Mediterranean cyclones are formed due to large scale forcing, i.e. the intrusion of upper tropospheric systems as a result of Rossby wave breaking over the Atlantic Ocean (Flaounas et al., 2022). This upper tropospheric forcing is identically introduced to the two simulations through the boundary conditions. However, cyclones seasonality and location also depend on diabatic forcing due to convection within the cyclone systems, as well as on the basin's orography. Both RCMs show a higher occurrence of cyclones in summer and spring (Fig. 2d and Fig. S1 in supplementary) and compared to ERA5, they tend to underestimate the CCD over the Mediterranean Sea while overestimating it over land and over the Aegean and Levantine Sea (Fig. 3b and c). Differences between the two RCMs and ERA5 arise primarily from the different resolution, dynamics and physical parameterisation. These factors influence how the models reproduce key processes, such as. the impact of orography on cyclone dynamics and the role of convection in deepening the cyclones, resulting in local deeper minima of SLP over Mediterranean areas with complex land-sea distribution. Therefore, differences from ERA5 should not be taken purely as a weakness of RCMs, but rather as a result of differences when reproducing atmospheric processes. Indeed, the magnitude of these*

*differences is comparable to the one found in previous studies (Flaounas et al., 2018a; Reale et al., 2022) and thus RCMs should be expected to deviate from reanalysis. In contrast, changes in the SST distribution have a minor impact on the dynamics of the cyclones, leading to small differences between STD and CPL, primarily in the location of cyclone minima over the sea (Fig. 3d). In conclusion, cyclone systems arise from a combination of large-scale processes (external to the cyclone) and small-scale processes (internal to the cyclone). In this context, atmosphere-ocean coupling is expected to have a stronger influence on the physical processes within the cyclone systems, and a rather weaker effect on their formation, distribution, and track characteristics."*

[Figure]

**Figure 3: Number of occurrences of cyclone centre densities (CCD) for the 500 most intense cyclones in ERA5 (a), along with CCD differences between STD and ERA5 (b), CPL and ERA5 (c), and CPL and STD (d). To highlight the cyclones' area of influence, each centre is represented by a circular area with radius of 1.5 degrees around the tracked minimum SLP point.**

**2 The spatial composite maps (Figs. 4-8) seem to be composites of time steps when one or more "common" selected intense cyclones are located within the domain of interest (Fig. 1). If this is the case, signals in these composite maps do not necessarily occur in the vicinity of intense cyclones of interest. This discrepancy could partly explain the study's apparently inconsistent conclusions (as noted above). While signals in composite maps during intense cyclone time-steps are associated with cyclone dynamics, some may not be directly relevant to cyclones.**

We thank the Reviewer for the comment. Regarding the inconsistent conclusions please refer to the comment right above. We also agree that to some extent, the differences fields in Figs 4-8 might be also due to differences in the larger scale atmospheric circulation, encompassing the cyclone systems. To address this issue, we show the composite field differences computed only within the area of influence of the cyclones, defined as a circular area with 500 km radius (Flaounas et al. 2016) around the minimum SLP tracking point (Fig. A1). These results are compared with the differences computed over the entire domain (as figure 7 in the revised manuscript but without bootstrapping for its significance, Fig. A2). Spatial distribution of differences in convective (Fig. A1e, A2e) and total (Fig. A1f, A2f) precipitation differences is fairly similar

between Fig. A1 and A2, suggesting that the impact of the SST distribution is stronger close to the cyclone centres. In addition, differences in atmospheric boundary layer processes, such as stronger latent (Fig. A2a) and sensible (Fig. A2b) heat fluxes, higher 10 m wind speed (Fig. A2c) and reduced stability (lower potential temperature lapse rate, Fig. A2d), are not well observed when computed only within the circular area around the cyclones (Fig. A1a-d). This reflects the small location and timing mismatches between the extreme winter cyclones in common between CPL and STD, i.e. when the SLP minima is within 500 km maximum distance and 12-hour time window. To highlight these outcomes, we added the following sentences in the manuscript (L197-203):

*"Our composite averaging is done for the entire domain and therefore the difference fields (CPL – STD) might be also affected by atmospheric systems other than cyclones. An additional analysis, using the same approach as in Flaounas et al., (2016), is applied where differences were calculated only within an area of 500 km around the cyclone centre. The different methods do not affect the results (not shown), because the intense cyclones are expected to have a substantial impact to the whole domain, so most of the differences are attributed to the areas close to cyclones. In addition, our strategy allows to overcome the slight location mismatch between CPL and STD (i.e. linked with 500 km maximum distance between the minimum of SLP) when computing the differences."*

[Figure]

**Figure A1:** Same as figure 7 but computed only on the area of influence of the cyclones, defined as a circular area with 500 km radius around the minimum SLP tracking point.

[Figure]

**Figure A2: Same as figure 7 but without applying the bootstrapping method for its significance.**

**3 The reasoning behind the differences in cyclone frequencies between RCMs and ERA5 (Figs. 2 and 3) is not quite convincing. The authors argue that the differences are due to the different native resolutions of the RCMs (higher) and the ERA5 model (lower). I am afraid, however, that ERA5 is a reanalysis assimilated with a lot of observations. Those observations are particularly abundant in the land and expected to substantially improve the reproducibility of synoptic and larger-scale features, including intense cyclones near the coast. It would make more sense to attribute differences in the open ocean mainly to models' native resolutions, but this is not the case for cyclones, for example, in the Adriatic Sea, the Ligurian Sea, and the Aegean Sea, and off the Gulf of Antalya. For synoptic-scale features, I think that reanalyses generally to a considerable extent reflect the reality, whose "native resolution" is much higher than any simulations.**

We agree with the Reviewer that ERA5 generally reflects real-world conditions, enhancing the reproducibility of large-scale synoptic systems such as Mediterranean cyclones. Therefore, in the revised

section 3.1, we have clarified that the RCMs do not simulate the impact of orography on cyclone dynamics better than ERA5, rather they show discrepancies compared to the reanalysis. These differences arise from the dynamical downscaling process, which differently simulates cyclone dynamics and the influence of orography on cyclone locations. Please see the discussion in the first specific comment and refer to the revised text (lines 247-271) mentioned there.

**4 The interpretation and discussion of SST differences (or biases) between STD and CPL are insufficient, and the authors should discuss the SST bias in relation to cyclone frequency and other properties. Is it related to the difference in the frequency and other properties of intense cyclones? Because there is no corresponding map of cyclone frequency difference to Figs. 5 and 6, it is hard to consider the relationship between the cyclone frequency and SST. To me, the SST difference (or bias) of nearly 2K is large enough to be expected to have impacts on the overlying atmosphere and cyclones (as in the following subsection). If SST differences indeed influence the atmosphere, it would be beneficial to clarify (or at least suggest) the mechanisms within the model that generate these differences.**

We thank the Reviewer for providing this valuable feedback. We agree that would be insightful to deeper investigate the SST bias in the CPL model and the physical mechanism behind that. Likely, the SST difference depends on the different forcings coming from the atmospheric models and intrinsic differences in the ocean models. This is an area of ongoing research in the climate modelling laboratory at ENEA, where the coupled model has been developed. However, this topic requires additional study and is beyond the scope of the present paper.

For the purposes of our work, it is important to investigate how the climatological differences between the explicitly resolved SST in CPL and the SST in STD (forced by ERA5) impact the atmospheric fields associated to the extreme cyclones. As discussed in Section 3.3 (revised manuscript), the SST differences are not linked to the cyclone's activity (frequency, timing and location). In fact, Figure 5 shows that the SST differences between CPL and STD during extreme winter cyclones (Fig. 5b) primarily reflect the climatological bias of the CPL model relative to observations (Fig. 5c).

**5 Related comment: Evaporation (or latent + sensible heat fluxes) are influenced by wind speed, surface temperature and specific humidity, and SST, following the bulk formula. Thus the evaporation difference (Fig. 7a) can result from the other atmospheric differences (e.g., Figs. 7b-c), rather than a one-way causation as in this study. While there is spatial alignment between SST bias and evaporation differences, this is not adequately addressed in the manuscript.**

We thank the Reviewer for the valuable feedback. Please note that we have changed figure 7 in the revised manuscript which now include latent and sensible heat fluxes instead of evaporation, following the suggestion of Reviewer 3. We agree that there is a mutual relation among the heat fluxes (evaporation), with both the SST, the specific humidity and the 10 m wind speed. We better clarify this in the revised paper (L342-347) as follows:

"*The warmer SST in the CPL model fosters latent and sensible heat fluxes at the sea surface (Fig. 7a, b), leading to increased vertical exchange of heat and moisture with the atmosphere. The stronger surface fluxes in CPL increase the turbulence and so the vertical mixing in the PBL, with warm air rising and cold air sinking due to buoyancy forces, transferring energy downward to the surface (downward momentum mixing, Hayes et al., 1989; Wallace et al., 1989), thus increasing the 10 m wind speed (Fig. 7c). The mutual relation among SST, surface fluxes and 10 m wind speed are confirmed by high Pearson correlation coefficients between the model differences (Fig. 9).*"

**6 Section 2.1 mentions that "STD" is an atmosphere-only WRF simulation with prescribed SST, while "CPL" is the ENEA-REG regional Earth system model including ocean, land, and freshwater fluxes and river discharge model. However, the authors argue that the only difference between STD and CPL resides in the SST over the Mediterranean Sea. Is this accurate? I am wondering if there is any significant difference on the land between the two simulations.**

We thank the Reviewer to point out this shortcoming. The only difference between the STD and the atmospheric component of CPL resides in the SST over the Mediterranean Sea. In the CPL simulation, the ocean component (i.e. MITgcm) exchanges only the SST field with the atmosphere, while STD uses the SST from ERA5. Therefore, the processes on land surface are simulated by WRF with identical settings between CPL and STD. In fact, over land, both CPL and STD present the same land scheme, Noah-MP and the same parameterizations. Please find below a scheme of the CPL model.

[Figure]

**7 L172: Given that cyclones are more intense in winter and the role of the SST and the air-sea fluxes on extreme events is expected to be stronger in autumn (LL106-107), it is counterintuitive to see the number of the selected most intense cyclones in winter and autumn is 199, while 142 (341-199) in summer and spring, corresponding to 71 cyclones in three months (JJA or MAM) on average. Does this mean there are fewer intense cyclones around the Mediterranean in autumn than in spring or summer? If so, I wonder why the focus of this study is only on autumn and winter Mediterranean cyclones.**

We thank the Reviewer for the valuable feedback. First, we would like to point out that there was a typo error in the manuscript: the same extreme cyclones between CPL and STD are 312 and not 341.

The seasonal cycle of intense Mediterranean cyclones is the follow: Winter (DJF) is the season with the higher frequency of intense cyclones, followed then by spring (MAM), autumn (SON) and summer (JJA). Among the 500 most intense cyclones, 312 are in common between CPL and STD: 129 in winter (DJF), 110 in spring (MAM), 17 in summer (JJA) and 56 in autumn (SON). We clarified this on the manuscript, and we added the reference to our methodology as follow (L176-184):

"*To compare CPL with STD in terms of sub-daily fields associated to the cyclones, the same events between the two simulations are selected. Two cyclones are considered the same event if their minimum of SLP is within a 500 km distance and within a time range of 12 hours. With these criteria, a total of 312 cyclones from the 500 most intense (around 62 %) are found in common between CPL and STD, of which 129 occurring in winter (DJF), 110 in spring (MAM), 17 in summer (JJA) and 56 in autumn (SON). This well align with results from Flaounas et al. (2018a), who also found that approximately 60% of the 500 most intense cyclone tracks were consistent between the coupled and standalone RCMs, using similar identification criteria Extending the distance criterion to 1000 km (i.e. the maximum area of influence of Mediterranean cyclones, Flaounas et al. 2016) and the time window to 48-hour, the percentage of detected cyclones in common between STD and CPL increases to 92%, but the outcomes of this study do not change (not shown).*"

As correctly pointed out by the reviewer, in our analysis we choose to investigate two seasons: DJF and SON. We have better explained why we chose these seasons in the introduction (L107-110) as follow:

"*For a more comprehensive analysis, two seasons are considered: the winter (DJF) when the cyclones are more intense and frequent (Campins et al., 2011; Flaounas et al., 2022) and autumn (SON) when the role of the SST and the air-sea fluxes on extreme events is expected to be stronger (Miglietta et al., 2011a; Ricchi et al., 2017). The enhanced surface fluxes in autumn result from the combination of relatively high SSTs, which are near their annual peak, and upper-level cold-air intrusions.*"

The winter cyclones are the main focus of the atmospheric analysis in section 3.2, while the analysis in SON can be found in the supplementary material. We have modified in section 3.2.3 the description of the autumn results, to highlight the differences with winter, as follow (L366-377):

"*The methodology used for winter is also applied to the 56 extreme autumn (SON) cyclones in common between CPL and STD. The SST differences between CPL and STD affect the atmospheric surface processes and PBL stability as seen in DJF, but with an opposite sign (Fig. S5), since in SON the CPL result colder (and not warmer as in DJF) than STD over most of the Mediterranean Sea (Fig. 6). Interestingly in SON, the intensity of surface heat fluxes (Fig. S3a, b) and precipitation (Fig. S3e, f) associated to extreme cyclones is even stronger than in DJF. The strong temperature gradient between warm Mediterranean Sea and cold atmospheric intrusions during SON cyclones reflects the amount of energy transferred to the atmosphere, amplifying precipitation intensity (Miglietta et al., 2011). Despite this, the differences between CPL and STD in cyclone-associated precipitation and 10 m wind speed (Fig. S5c, e and f) are non-statistically significant, and less correlated with the SST differences (Fig. S6). This may be partially attributed to the smaller SST differences (Fig. 6 vs. Fig. 5) over the Balearic and Tyrrhenian Seas, where most SON extreme cyclones occur (Fig. S1). The strong impact of the SST distribution and air-sea fluxes on the atmosphere is expected to be significant on specific autumn events, as already shown and discussed in previous studies (Akhtar et al., 2014; Berthou et al., 2015, 2016; Miglietta et al., 2011; Ricchi et al., 2017).*"

For completeness, we also analysed the spring (MAM) season (see below), but we have decided to don't add this analysis to the manuscript since it does not add any additional insights compared to the DJF case. In fact, the differences in SST between CPL and STD are substantially reduced compared to DJF (Fig. A4 versus Fig. 5), especially over the gulf of Genoa and the gulf of Lyon, where most of the intense cyclones are located (Fig. S1 in supplementary). Moreover, the surface heat fluxes (Fig. A3a, b) and the cyclones' associated precipitation (Fig. A3e, f) and wind speed (Fig. A3c) in spring are less intense than in winter (Fig 4). So, both the lower SST differences and the weaker atmospheric processes associated to the cyclones lead to less intense and statistically non-significant differences in convective (Fig. A5e) and total precipitation (Fig. A5f, same bootstrapping method applied for DJF) and lower linear correlation between the SST and the atmospheric fields differences (Fig. A6). This result confirms our conclusion, i.e. the different SST distribution between CPL and STD is the dominant factor in shaping the differences in sea surface fluxes, atmospheric stability, 10 m wind speed and precipitation associated to the extreme cyclones.

[Figure]

**Figure A3: Same as figure 4 but for MAM.**

[Figure]

**Figure A4: Same as figure 5 but for MAM.**

[Figure]

**Figure A5: Same as figure 7 but for MAM.**

[Figure]

**Figure A6: Same as figure 9 but for MAM.**

**Minor comments:**

1. **L15: "in the second WRF" -> "in the second simulation the WRF" ?**

   It has been corrected.

2. **L18: similarly reproduced -> "similar" ?**

   It has been corrected.

3. **L19: regardless -> regardless of**

   It has been corrected.

4. **L21: planet -> planetary**

   It has been corrected.

5. **L32: "Sea Surface Temperature" -> "sea surface temperature"**

   It has been corrected.

6. **L35: "midlatitude cyclones, entering the Mediterranean basin" -> "midlatitude cyclones entering the Mediterranean basin" ?**

It has been corrected.

7. **L64: long ago -> long**

   It has been corrected.

8. **L67 "boundary conditions, that becomes" -> "boundary conditions, which becomes"**

   It has been corrected.

9. **L69: challenged -> attempted?**

   It has been corrected.

10. **L87: "improves the track length" -> "improves the reproducibility of the track length" ?**

    It has been corrected.

11. **L95: investigating -> by investigating**

    It has been corrected.

12. **L95: affects -> affect**

    It has been corrected.

13. **L103: "the explicitly resolved SST" -> "do the explicitly resolved SST"?**

    We have modified the second research question as follow (L103-105):

    "*How do differences in SST distribution shape atmospheric processes within the planetary boundary layer (PBL) during extreme cyclone events, and how do these mechanisms, in turn, impact the cyclone-related precipitation and wind speed?*".

14. **L107: I am not sure why the role of the SST and the air-sea fluxes on extreme events is expected to be stronger in autumn than in winter.**

    We have better explained in the introduction (L107-110) why the air-sea fluxes are stronger during the autumn extreme cyclones with specific references:

    "*For a more comprehensive analysis, two seasons are considered: the winter (DJF) when the cyclones are more intense and frequent (Campins et al., 2011; Flaounas et al., 2022) and autumn (SON) when the role of the SST and the air-sea fluxes on extreme events is expected to be stronger (Miglietta et al., 2011a; Ricchi et al., 2017). The enhanced surface fluxes in autumn result from the combination of relatively high SSTs, which are near their annual peak, and upper-level cold-air intrusions.*"

15. **L108: "next" -> "the next"**

    It has been corrected.

16. **L118: "extensively used" -> "which is extensively used"**

    It has been corrected.

17. **L123: I think that 1/12deg is substantially less than approximately 10km, especially in the zonal direction.**

    The reviewer is right, as the grid is irregular in some points the nominal resolution of 1/12° corresponds to less than 10 km. We have removed "(approximately 10 km)" from the sentence.

18. **L124 (the Med-CORDEX region): It would be helpful to refer to Fig. 1 here.**

Thanks for the suggestion, we have added "(Fig. 1)" in line 133.

19. **L125: The resolution of the SST prescribed to ERA5 depends on the period. In particular, the resolution is substantially different between HadISST2 (~2007) and OSTIA (2007~). The description of ERA5 SST as Δx~0.25deg might be misleading.**

The reviewer is correct, SST, in ERA5, is given by two external providers with two different nominal resolutions. Before September 2007, SST from the HadISST2 dataset (Δx = 0.25deg) is used, and from September 2007 onwards, the OSTIA (Δx = 0.05deg) dataset is used. However, the SST is provided by the Copernicus Climate Data Store, for the whole period, at 0.25deg horizontal resolution. To be more precise, we have modified the text (123-129) as follow:

"*Thus, the only difference between the STD and the CPL simulation resides in the SST over the Mediterranean Sea, which derives from the ERA5 SST reanalysis (daily, Δx = 0.25°) in STD, whereas it comes interactively from MITgcm (3-hourly, Δx = 1/12°) in CPL. To note that SST, in ERA5, is given by two external providers with two different nominal resolutions. Before September 2007, SST from the HadISST2 dataset (Δx = 0.25deg, Titchner and Rayner, 2014) is used, and from September 2007 onwards, the OSTIA (Δx = 0.05deg, Donlon et al., 2012) dataset is used. However, the Copernicus Climate Data Store provides the SST filed at 0.25° horizontal resolution for the whole period.*"

20. **L146: "cyclone tracking algorithm" might be preferable to "storm track method."**

We have changed the title of section 2.2.1 in: "Storm track method" as suggested.

21. **L147: "CYCLOYTRACK" -> "CycloTRACK" (see Flaounas et al. 2014)**

It has been corrected.

22. **L147: "Mean Sea Level Pressure" -> "mean sea level pressure"**

It has been corrected.

23. **L157: Referring to terrain > 800m as "high mountain environment" sounds a bit weird.**

The reviewer is correct, and we have changed the text (L166-168) accordingly:

"*A terrain filter of 800 m altitude has been also applied to focus on the intense cyclones over the sea and second to filter out algorithm artefacts, that tend to form over mountains due to the extrapolation of pressure fields on sea level (Neu et al., 2013).*"

24. **L163: The minimum SLP during a cyclone's lifetime depends on the background, larger-scale SLP distribution than the synoptic scale. Is the climatological-mean SLP nearly uniform in the Mediterranean Sea?**

We thank the reviewer for the interesting question. As discussed in the first specific comment, a cyclone intensity (i.e. defined here as a SLP local minimum) mostly depends on large scale forcing (i.e. baroclinic forcing) and on diabatic forcing (convection close to the cyclone centre). As shown in Flaounas et al., (2021), the relative contribution of these two forcings is highly dependent on the case and therefore on the cyclones' proximity to mountain volumes, the underlying SST, the season etc. The Reviewer's suggestion is interesting, and we have addressed it by computing the SLP field during the winter and autumn extreme cyclones (figures A7, A8, A9, A10 below). Both in winter (Fig. A7) and in autumn (Fig. A8), the mean SLP distribution during the extreme cyclones of the CPL simulation (same results for STD, not shown) is coherent with the distribution of the frequency and location of the cyclones (Fig. S1 in supplementary). In addition, when looking at the differences between CPL and STD, the SLP is affected by the SST values. In DJF (Fig. A9) the SST of the CPL is warmer and the SLP is lower (deeper minima of the cyclones), while in SON (A10) the SST is colder and the SLP is higher. This could be related to the pressure adjustment mechanism (Lindzen

and Nigam, 1987): warmer SSTs induce thermal expansion of the air lowering the atmospheric pressure at the surface.

[Figure]

**Figure A7: Map for sea-level pressure (SLP) from the CPL simulation during winter extreme cyclones in common with CPL.**

[Figure]

**Figure A8: Same as figure A7 but for SON.**

[Figure]

**Figure A9: Map of the differences between CPL and STD during the common extreme winter cyclones for SLP.**

[Figure]

**Figure A10: Same as figure A10 but for SON.**

**25  L172: Please specify what "DJF" and "SON" stand for here.**
It has been corrected.

**26  L179: STD e CLP -> STD and CPL**
It has been corrected.

**27  L182-183: The authors argue that "the influence of cyclones on the atmospheric state is independent of the location of the cyclones in the Mediterranean Sea". Why?**

We thank the Reviewer for pointing out the need for clarification. This sentence (L182-183) was not clear, so, we removed it and change the text as follow (L197-203):

"*Our composite averaging is done for the entire domain and therefore the difference fields (CPL – STD) might be also affected by atmospheric systems other than cyclones. An additional analysis, using the same approach as in Flaounas et al. (2016), is applied where differences were calculated only within an area of 500 km around the cyclone centre. The different methods do not affect the results (not shown), because the intense cyclones are expected to have a substantial impact to the whole domain, so most of the differences are attributed to the areas close to cyclones. In addition, our strategy allows to overcome the slight location mismatch between CPL and STD (i.e. linked with 500 km maximum distance between the minimum of SLP) when computing the differences.*"

**28 L229: The present study utilizes the same cyclone tracking algorithm as Flaounas et al. (2023). Thus the authors should specify references for "different cyclone tracking methods" here.**

Mentioning "different cyclone tracking methods", we probably created a misunderstanding. Here we apply a single cyclone tracking method to our datasets and thus it would be rather odd to include -a plethora of- additional references for methods that we do not use. We revised the text as follows (L246-247):

"*These results are in fair agreement with the most intense cyclones in ERA5 as defined by composite reference tracks for the Mediterranean (Flaounas et al., 2023). "*

**29 3: Consider using a more realistic mask shape (e.g., circles) over 3degx3deg square domains. Additionally, which season(s) is this figure showing? DJF+SON?**

We changed figure 3 by using a circle area with 1.5 deg. radius around the minima of SLP tracking points. Please see the revised figure 3.

**30 L256-259: I consider that the explanation of the structure of section 3.2 (subsection) here is not needed.**

We thank the reviewer to point out this shortcoming. We have modified the structure of section 3 as follow:

3.1 Climatology of extreme Mediterranean cyclones

3.2 Atmospheric fields during extreme cyclones

3.3 SST differences between CPL and STD

3.4 Impact of the SST distribution on cyclones' precipitation

3.5 Ocean response to extreme cyclones

**31 L264: It does not make sense to me that cyclones do not turn into precipitation. (Additionally, the sentence here seems grammatically incorrect.)**

The reviewer is correct, the sentence was not clear. We changed the sentence as follow (L289-291):

"*This precipitation pattern is associated with winter cyclones generally coming from the west, as indicated by Flaounas et al. (2015) and Raveh-Rubin and Flaounas (2017) and interacting with the complex orography of the basin, increasing precipitation over coastal areas.*"

**32 "PBL is higher" -> "PBL height is larger"?**

Yes, we meant that the height of the PBL is larger. We have corrected it.

**33  4: The panel labels are hard to find. Please improve their visibility.**

We have increased the size of the panel labels in all the figures in the revised manuscript.

**34  If possible, the results in Fig. 4 should be compared with observations (e.g., ERA5).**

We thank the Reviewer for the suggestion. Following the first general comment of the Reviewer 3, we have performed the comparison of the atmospheric fields during the 500 most intense cyclones in ERA5, STD and CPL (Fig. A11, A12, A13 below). In figure A11 we show the distribution of the fields during the 500 most intense cyclones in ERA5, while in figures A12 and A13, we show the differences of STD and CPL with ERA5, respectively. The RCMs tend to underestimate the surface heat fluxes over the sea (Fig A12a, b; Fig. A13a, b), to overestimate the sensible heat over land, especially in the north Africa region (Fig A12b; Fig. A13b), and to present a slightly higher stability of the PBL (higher potential temperature lapse rate) over the south and east Mediterranean Sea (Fig A12d; Fig. A13d). The surface wind speed is higher for the RCMs in most regions (Fig A12c; Fig. A13c), likely due to the higher resolution and different physical parametrisation in WRF, while the convective precipitation is underestimated by RCMs especially over the coastal area. Finally, looking into the total (convective and large-scale) precipitation differences, the RCMs simulate a stronger precipitation over the sea, while over land tend to overestimate it in mountainous regions (Alps, Pyrenees and Greek and Turkish mountains) and underestimate it on the west coasts of Italy and Balkans.

This analysis has not been included in the manuscript because the focus of the study is not to validate the RCMs against ERA5, since is already done by Anav et al. (2024). Instead, our paper investigates how the atmosphere-ocean coupling, resulting in a differing SST distribution between CPL and STD configuration, influences the key atmospheric processes associated with extreme cyclones. We clarified this in the revised manuscript (section 2.2, "storm track method"), as follow (L153-156):

*"A storm track method is applied to both ERA5 reanalysis and RCM simulations. To note that the comparison of the models with ERA5 is restricted to the evaluation of the RCMs' ability to reproduce the climatology of the extreme cyclones, in terms of their seasonal cycle, track characteristics and spatial distribution. In fact, the full evaluation of the RCMs against ERA5 was already performed by Anav et al. (2024)."*

[Figure]

**Figure A11: Maps for latent heat flux (a), sensible heat flux (b), 10 m wind speed (c), potential temperature lapse rate (d), convective precipitation (e) and total (large-scale + convective) precipitation (f) from ERA5 during the 500 most intense cyclones.**

[Figure]

**Figure A12: Maps of the differences between STD and ERA5 during the 500 most intense cyclones for latent heat flux (a), sensible heat flux (b), 10 m wind speed (c), potential temperature lapse rate (d), convective precipitation (e) and total precipitation (f).**

[Figure]

**Figure A13: Same as figure A12 but for CPL.**

**35  L277: The title "SST analysis" is obscured and needs to be clarified.**

We changed the title in, "*SST differences between CPL and STD*"

**36  The figures' sequence (including supplementary ones) could be improved to follow numerical order.**

We have changed most of the figures in both the revised manuscript and supplementary material and they all follow the same sequence of the atmospheric fields.

**37  L288 "All the outcomes on DJF are also valid for the analysis of the SST bias in SON (Fig. S6)": It seems contradictory with the following statement. Please clarify which outcomes in DJF are valid for SST bias analysis in SON.**

The Reviewer is correct. We have changed the sentence in the revised manuscript (L367-369) as follow:

"*The SST differences between CPL and STD affect the atmospheric surface processes and PBL stability as seen in DJF, but with an opposite sign (Fig. S3 in supplementary), since in SON the CPL result colder (and not warmer as in DJF) than STD over most of the Mediterranean Sea (Fig. 6)*."

**38  L288 Fig. S6 -> Fig. 6**

It has been corrected.

**39  5-8: White contours for significance are difficult to identify. Consider improving visibility. (Perhaps There are a lot of white contours to white out insignificant differences with colors?)**

We thank the reviewer for pointing out this issue. We have tested alternative configurations to improve the visibility of statistically significant points, such as using black dots to mark them (see Fig. A14 below). However, adding these markers overloaded the plots with elements, obscuring the colour differences, which are essential for interpretation. We chose white contours to represent significance because there are no statistically significant grid points with values close to zero. So, the white colours almost everywhere indicate non-significant grid points and not differences close to zero, ensuring clarity in the interpretation of the plots.

[Figure]

**Figure A14: Same as figure 7 but with black markers for significance.**

**40  L311: What correlation is "the high correlation" (and Fig. 9)? Spatial correlation over the sea?**

With high linear correlation we mean a Pearson correlation coefficient greater than 0.7. The correlations among the differences are computed both over land and sea and for only the statistically significant grid points of both fields.

**41  L315 Clarify if the observed link between warmer SST and higher 10 m wind speed is related to the "vertical mixing effect" from Wallace et al. (1989) and Hayes et al. (1989).**

The reviewer is correct, the link between warmer SST and higher 10 m wind speed is indeed related to the downward momentum mixing (DMM, Wallace et al. 1989; Hayes et al. 1989). We have clarified this as follow (L343-347):

*"The stronger surface fluxes in CPL increase the turbulence and so the vertical mixing in the PBL, with warm air rising and cold air sinking due to buoyancy forces, transferring energy downward to the surface (downward momentum mixing, Hayes et al., 1989; Wallace et al., 1989), thus increasing*

*the 10 m wind speed (Fig. 7c). The mutual relation among SST, surface fluxes and 10 m wind speed are confirmed by high Pearson correlation coefficients between the model differences (Fig. 9).*"

**42   L319: I could not follow the argument that "the stronger horizontal winds in CPL lead to a mismatch between areas of high vertical moisture flux and total precipitation". What is "high vertical moisture flux" and how is the mismatch related to the stronger horizontal winds? This paragraph (L319-326) should be improved so as to be understood more easily.**

We thank the reviewer to point out this shortcoming. We have changed the paragraph (L358-360) to make it clearer:

"*The total (large-scale and convective) precipitation differences between the models result not only from direct changes in the surface fluxes but also from the wind dynamics that are responsible to the changes in the convergence zones of moisture, as discussed in Berthou et al. (2016).*"

**43   L 389: Please clarify the term "mean cooling."**

With "mean cooling" we meant the cooling effect averaged over the number of the cyclones. We have decided to just say "cooling" in the sentences to don't create confusion.

**44   Related comment to major comment (B): I think that showing the composited wind (Fig. 4d) is misleading, because the wind is the superposition of all the events considered, which come mainly from those around Italy but include those occurring in distant regions (Fig. 3). In other words, the precipitation distribution (Figs. 4a-b) does not necessarily occur associated with the wind pattern in Fig. 4d.**

The Reviewer is correct, on each cyclone event with specific precipitation distribution, the wind speed intensity and direction are locally different from the composite wind shown in figure 4c (revised manuscript). However, our priority is to show the climatological mean of the composite fields during the extreme cyclones, so that we can compare them between the two simulations. Therefore, the composite fields in figure 4, reflect their mean distribution over the effective area of high cyclones' frequency, which is mainly concentrated over the central Mediterranean basin. This includes high precipitation along the coasts of Italy and the Balkans (Fig 4f), as well as strong winds over the Sea associated with cyclonic circulation (Fig4c).

---

## Author Comment (AC3)

**Reviewer #3:**

**This paper aims to quantify the impact of air-sea interactions during extreme cyclone events on the structure of the atmospheric and oceanic boundary layers. This is addressed by performing two 33-year simulations using coupled (CPL) and atmosphere only (STD) models. First the cyclone climatologies from the CPL and STD simulations are compared to an atmospheric reanalysis dataset (ERA5). Then the climatological SST fields during extreme cyclones from the CPL and STD are compared to satellite-based SST dataset (MED-REP-L4). Next, the CPL and STD atmospheric fields are compared and finally the evolution of the ocean structure in the CPL simulation is compared to and ocean reanalysis dataset (CMES). I found the paper a bit confusing to read and the grammar is incorrect in many places. The motivation for the analysis and importance of the study needs more emphasis before the paper can be considered to be suitable for publication.**

We thank the Reviewer for dedicating time to review our manuscript and for the detailed observations that have raised the quality of the paper.

The valuable feedback has helped us improve the clarity of the work and better emphasise its results and novelty. Specifically, we have significantly revised the Results section to provide a more detailed explanation of the physical mechanisms behind the differences in the atmospheric processes between CPL and STD during extreme Mediterranean cyclones and to address the specific questions raised by the reviewer.

In addition, we have revised the conclusion to emphasize the novelty of the study. Specifically, we modified lines 432-434, as follow:

"*This study investigates for the first time (to the best of our knowledge) how extreme Mediterranean cyclones affects simultaneously the atmosphere and the ocean at different vertical levels, comparing two high-resolution RCM simulations, one atmosphere-ocean coupled (CPL) and one atmosphere stand-alone (STD), over the period 1982-2014.*",

and lines 454-456, as follow:

"*This research highlights the ability of the coupled model to coherently simulate the entire atmosphere-ocean system, thus providing new insights into how sea surface energy is redistributed between the atmospheric boundary layer and the ocean mixed layer, and how this impacts the precipitation and the wind speed during extreme cyclone events.*".

Please note that figures 4, 7 and 9 have been revised following the reviewers' comments. The figures now show latent and sensible heat fluxes, 10 m wind speed, potential temperature lapse rate, convective precipitation and total (large scale + convective) precipitation.

Please also note that the text-line references mentioned in our responses correspond to the revised manuscript.

Below, the Reviewer will find our detailed, point-by-point, answers.

**General comments:**

**1 There are fairly large differences between the CPL/STD cyclone climatologies and ERA5, shown in figs 2 & 3, which are dismissed as small in the paper. Do these differences result in differences in the cyclone-climatology atmospheric fields (precipitation, pbl height, 10m wind speed, evaporation and 2m specific humidity)? Figure 4 does not compare with ERA5 fields so it's not possible to determine the answer to this question.**

We thank the Reviewer for this comment. We agree that one may indeed argue that differences among the RCMs and ERA5 are fairly large in Figs 2&3. Nevertheless, differences do not reject the suggestion that "similar" (rather than "identical") spatial distributions, track characteristics and seasonal cycles are followed by all three datasets. This should be an expected result. As also discussed in our replies to Reviewers 1 and 2:

the most intense Mediterranean cyclones are formed due to large scale forcing, i.e. the intrusion of upper tropospheric systems as a result of Rossby wave breaking over the Atlantic Ocean (Flaounas et al., 2022). This upper tropospheric forcing is identically introduced to the two simulations through the boundary conditions. Furthermore, their characteristic length scale allows their realistic reproduction even at relatively coarse resolutions as the ones in ERA5. The development of cyclones though might depend on both the large scale forcing and diabatic forcing due to convection within the cyclone systems. The latter is strongly dependent on the parametrisation, resolution and -therefore- on the underlying SST. In such climate-scales numerical experiments, one should expect thus that cyclones formation should be rather "similar" in the two simulations, but cyclones development might change at the extent that an extreme cyclone is more diabatically driven than developed due to baroclinic instability (i.e. due to the upper level forcing which should be less sensitive to SST). As a conclusion, if we regard a cyclone system as the outcome of large-scale processes (external to the cyclone system) and small-scale processes (internal to the cyclone system), then we should expect atmosphere-ocean coupling to have a stronger effect on the physical processes of the cyclone systems, and a rather weaker effect on their formation distribution and track characteristics. We have modified Fig. 3 and included this discussion in the revised manuscript (section 3.1, "Climatology of extreme Mediterranean cyclones") as follow (L247-271):

"*Figure 3 instead shows the maps of cyclone centre densities (CCD; Neu et al. 2013, Flaounas et al. 2018a) for ERA5 and the differences in CCD between ERA5 and RCMs. The CCD is defined as the absolute number of occurrences of the 500 most intense cyclone centres. To highlight the cyclones' area of influence, each centre is represented by a circular area with radius of 1.5 degrees around the tracked minimum SLP point. Compared to ERA5, both RCMs tend to capture the main regions of frequent cyclogenesis (over the gulf of Genoa, over the Adriatic and Aegean Seas and the marine areas close to Cyprus). This can be expected since the most intense Mediterranean cyclones are formed due to large scale forcing, i.e. the intrusion of upper tropospheric systems as a result of Rossby wave breaking over the Atlantic Ocean (Flaounas et al., 2022). This upper tropospheric forcing is identically introduced to the two simulations through the boundary conditions. However, cyclones seasonality and location also depend on diabatic forcing due to convection within the cyclone systems, as well as on the basin's orography. Both RCMs show a higher occurrence of cyclones in summer and spring (Fig. 2d and Fig. S1 in supplementary) and compared to ERA5, they tend to underestimate the CCD over the Mediterranean Sea while overestimating it over land and over the Aegean and Levantine Sea (Fig. 3b and c). Differences between the two RCMs and ERA5 arise primarily from the different resolution, dynamics and physical parameterisation. These factors influence how the models reproduce key processes, such as. the impact of orography on cyclone dynamics and the role of convection in deepening the cyclones, resulting in local deeper minima of SLP over Mediterranean areas with complex land-sea distribution. Therefore, differences from ERA5 should not be taken purely as a weakness of RCMs, but rather as a result of differences when reproducing atmospheric processes. Indeed, the magnitude of these differences is comparable to the one found in previous studies (Flaounas et al., 2018a; Reale et al., 2022) and thus RCMs should be expected to deviate from reanalysis. In contrast, changes in the SST distribution have a minor impact on the dynamics of the cyclones, leading to small differences between STD and CPL, primarily in the location of cyclone minima over the sea (Fig. 3d). In conclusion, cyclone systems arise from a combination of large-scale processes (external to the cyclone) and small-scale processes (internal to the cyclone). In this context, atmosphere-ocean coupling is expected to have a stronger influence on the physical processes within the cyclone systems, and a rather weaker effect on their formation, distribution, and track characteristics.*"

[Figure]

**Figure 3: Number of occurrences of cyclone centre densities (CCD) for the 500 most intense cyclones in ERA5 (a), along with CCD differences between STD and ERA5 (b), CPL and ERA5 (c), and CPL and STD (d). To highlight the cyclones' area of influence, each centre is represented by a circular area with radius of 1.5 degrees around the tracked minimum SLP point.**

In addition, we attached below the comparison of the atmospheric fields during the 500 most intense cyclones in ERA5, STD and CPL (Figs. A1, A2, A3). In figure A1 we show the distribution of the fields during the 500 most intense cyclones in ERA5, while in figures A2 and A3, we show the differences of STD and CPL with ERA5, respectively. The RCMs tend to underestimate the surface heat fluxes over the sea (Fig A2a, b; Fig. A3a, b), to overestimate the sensible heat over land, especially in the north Africa region (Fig A2b; Fig. A3b), and to present a slightly higher stability of the PBL (higher potential temperature lapse rate) over the south and east Mediterranean Sea (Fig A2d; Fig. A3d). The surface wind speed is higher for the RCMs in most regions (Fig A2c; Fig. A3c), likely due to the higher resolution and different physical parametrisation in WRF, while the convective precipitation is underestimated by RCMs especially over the coastal area. Finally, looking into the total (convective and large-scale) precipitation differences, the RCMs simulate a stronger precipitation over the sea, while over land tend to overestimate it in mountainous regions (Alps, Pyrenees and Greek and Turkish mountains) and underestimate it on the west coasts of Italy and Balkans.

This analysis has not been included in the manuscript because the focus of the study is not to validate the RCMs against ERA5, since is already done by Anav et al. (2024). Instead, our paper investigates how the atmosphere-ocean coupling, resulting in a differing SST distribution between CPL and STD configuration, influences the key atmospheric processes associated with extreme cyclones. We clarified this in the revised manuscript (section 2.2, "storm track method"), as follow (L153-156):

"*A storm track method is applied to both ERA5 reanalysis and RCM simulations. To note that the comparison of the models with ERA5 is restricted to the evaluation of the RCMs' ability to reproduce the climatology of the extreme cyclones, in terms of their seasonal cycle, track characteristics and spatial distribution. In fact, the full evaluation of the RCMs against ERA5 was already performed by Anav et al. (2024).*"

[Figure]

**Figure A1: Maps for latent heat flux (a), sensible heat flux (b), 10 m wind speed (c), potential temperature lapse rate (d), convective precipitation (e) and total (large-scale + convective) precipitation (f) from ERA5 during the 500 most intense cyclones.**

[Figure]

**Figure A2: Maps of the differences between STD and ERA5 during the 500 most intense cyclones for latent heat flux (a), sensible heat flux (b), 10 m wind speed (c), potential temperature lapse rate (d), convective precipitation (e) and total precipitation (f).**

[Figure]

**Figure A3: Same as figure A2 but for CPL.**

**2 There are large differences between CPL and satellite-based SST cyclone climatology fields (0.5K) shown in figs 5 and 6. Unsurprisingly, these differences go on to dominate the spatial maps shown in figures 7 and 8, as demonstrated by the high correlations in figure 9. I was missed the importance of this result. The authors have demonstrated that differences in SSTs leads to large differences in the atmospheric fields, but what is the link to the extreme cyclones. Are the SST differences larger during cyclone events than non-cyclone events, and thus accurate prediction of extreme cyclones is important? If so, do the noncyclone climatologies also need to be included to demonstrate this?**

We thank the Reviewer for this thoughtful comment. Figure 5b and c show respectively the SST difference between CPL and STD during the winter extreme cyclones and on climatological scale. The two figures are very similar, thus indicating that the SST differences during cyclone events are not larger than the climatological differences. However, this significant climatological SST difference has a greater impact on the atmospheric processes during extreme cyclones than on climatological scale, due to enhanced air-sea exchange

of energy, wind speed and convection during cyclonic events (Fig. S2 below). The new Figure S2, added in the supplementary material, shows the atmospheric fields when considering only extreme cyclones or the climatological scale in winter (DJF) for STD. Similar results are also found for CPL (not shown).

In response to the reviewer's question, we also added comments in section 3.2 of the revised manuscript (L298-305) as follow:

"*It is interesting to note that, in the winter climatology, the total precipitation are much smaller compared to cyclone events. This can be explained by the intense baroclinic forcing during winter cyclones that trigger convection and intensify the winds at the surface, enhancing the transfer of energy from the sea to the atmosphere and thus increasing the vertical transport of heat and moisture. Figure S2 in supplementary shows the differences between cyclones composite fields and climatological fields in winter for STD (same results for CPL, not shown), where is clear the higher latent heat, (Fig. S2a), sensible heat (Fig. S2b) and 10 m wind speed, the lower stability (S2d) and the stronger precipitation (Fig. S2e, f) in the areas of cyclones' locations. This highlights the greater importance of the Mediterranean SST as source of energy for the cyclones when the air-sea exchange processes are stronger, with intense precipitation and wind speed.*"

[Figure]

**Figure S2: Maps of the differences in latent heat flux (a), sensible heat flux (b), 10 m wind speed (c), potential temperature lapse rate (d), convective precipitation (e) and total (large-scale + convective) precipitation (f) between cyclones and climatological scales for STD in winter (DJF).**

**3 Figure 10 is the most interesting result because it removes the removes the bias in SST and thus allows a comparison of the effect of coupling. What causes the difference between CMEMS and STD SST evolution (using daily ERA5 SST) prior to the cyclone event?**

We thank the reviewer to raise this interesting question. The CMEMS MED-Currents (Escudier et al. 2021) is a high-resolution (1/24°) Mediterranean Sea physical reanalysis, while the ERA5 uses two different SST dataset with different nominal resolutions, i.e. HadISST2 ($\Delta x = 0.25$deg, Titchner and Rayner, 2014) before September 2007 and OSTIA ($\Delta x = 0.05$deg, Donlon et al., 2012) afterwards. However, the Copernicus Climate Data Store provides the SST field only at 0.25° horizontal resolution for the whole period.

It is interesting to note that the CMEMS reanalysis is forced by atmospheric fields of ERA5. Thus, the different SST between CMEMS and ERA5 (SST in STD) is probably related to the different ocean model implemented,

resolution and assimilated observations. In the work of Escudier et al. 2021 they compare the CMEMS MED-Currents only with the previous version of the Mediterranean reanalysis and not with ERA5 SST. So, further research would be needed to investigate what causes the differences in SST between CMEMS and ERA5 reanalysis and this goes beyond the scope of the present paper.

**Specific comments:**

**1 Line 12, 13: ENEA-REG, Med-CORDEX, ERA5, WRF, MITgcm acronyms need to be defined. Is it important for the general reader to know the names of these datasets and models? If not, please consider writing the abstract using more general language and leave the detailed acronyms to the main body of the paper.**

We have revised the abstract to use more general language and avoid the use of acronyms.

**2 Line 24: What do the authors mean by the 'effectiveness' of the coupled model?**

We have corrected "effectiveness" with "ability in lines L24-26 as follow:

"*The analysis shows the ability of the coupled model to coherently represent the dynamic and thermodynamic processes associated with extreme cyclones across both the atmosphere and the ocean.*"

**3 Line 54: Which side of the Alps is the 'leeward side'? Surely, this depends on the wind direction?**

The Reviewer is correct. The leeward side of the mountains corresponds to the downwind side and so depend on the wind direction.

**4 Line 80: What do the authors mean by 'proper' air-sea coupling effects?**

By "proper" air-sea coupling effect, we refer to the influence of the coupling on atmospheric fields, specifically related to the direct exchange of information between the atmospheric and ocean models and not dependent on the impact of the different SST distribution on the atmosphere. For clarity, we modified the sentence as follow (L79-81):

"*Berthou et al. (2014, 2015, 2016) found that only a minor part of the change in precipitation was strictly due to the air-sea coupling effects, while the long-term difference in SST between the simulations was responsible for most of the change*"

**5 Line 107: Why is the role of SST and air-sea fluxes on extreme events expected to be stronger in the Autumn season?**

We have added in the introduction (L107-110) the physical reason and the references on why the air-sea fluxes are expected to be stronger in the autumn season, as follow:

"*For a more comprehensive analysis, two seasons are considered: the winter (DJF) when the cyclones are more intense and frequent (Campins et al., 2011; Flaounas et al., 2022) and autumn (SON) when the role of the SST and the air-sea fluxes on extreme events is expected to be stronger (Miglietta et al., 2011a; Ricchi et al., 2017). The enhanced surface fluxes in autumn result from the combination of relatively high SSTs, which are near their annual peak, and upper-level cold-air intrusions.*"

**6 Line 163: 500 cyclones represent almost 20% of the cyclone distribution. This does not seem particularly extreme.**

The reviewer is correct. From a statistical point of view, the 500 most intense cyclones do not represent the "extremes" of the distribution, but we also needed to guarantee to have enough cyclones at least in DJF and SON season. We simply use the term "extreme" as a way to refer to the "500 most intense" (in terms of minimum SLP) cyclones.

**7 Line 175: I found the terminology q-gradient ambiguous. Why not use static stability or potential temperature lapse rate which are more standard terms for such a metric?**

We have changed the terminology in "potential temperature lapse rate", as suggested by the reviewer.

**8 Line 216: Why is a radius of 1.5o around the cyclone centre chosen? This seems to suggest that the enhanced surface fluxes occur very close to the cyclone centre and are axisymmetric. Is this supported by any analysis?**

We thank the reviewer to point out this shortcoming. The total area of influence of the cyclones is on average larger than a circle with 1.5° radius. For instance, Flaounas et al. 2016 used a circular area with a radius that vary dynamically according to the relative vorticity field in the vicinity of the cyclones centre. They found that the median of all cyclones' effective area is of the order of 500 km, while their 5th and 95th quantile is of the order of 150 and 1050 km, respectively. In the ocean analysis (section 3.5 in the revised manuscript) we choose to use a smaller area of influence to have a greater amount of grid points over the Sea concentrated around the minima of the cyclones and not closed to the land and/or the islands where the ocean is too shallow. The distribution of the cyclone's minima with their effective area of influence is shown in supplementary in figure S7.

**9 Figure 2: This figure shows that there is some seasonal dependence on the performance of the CPL and STD model simulations when compared to ERA5 for predicting the number of cyclones. Is there also seasonal dependency in the other statistics (intensity, lifetime and speed)?**

The Reviewer is correct, the RCMs tend to overestimate the intense cyclones in spring and summer and underestimate them in winter compared to ERA5. However, the main statistics are not affected by this seasonality behaviour. For instance, in winter the RCMs present similar statistics compared to ERA5, as shown in figure A4 below, and similarly to Figure 2.

[Figure]

**Figure A4: Statistics, intensity (a), lifetime (b) and speed (c) of the extreme cyclones in winter (DJF) for STD, CPL and ERA5.**

**10 Figure 3: The maxima in frequency of cyclones in CPL model, over land, does not correspond to maxima in the STD simulation. Furthermore, the frequency over the ocean reduces in the CPL simulation. Therefore, I do not think the evidence supports the statement on line 230 that 'the spatial distribution of extreme cyclones is similarly reproduced by the models compared to ERA5' or on line 241 that 'differences between STD and CPL are limited and non-significant'. Perhaps difference plots would demonstrate the similarity or differences in the simulations more clearly?**

We thank the reviewer to raise this doubt. We have changed the paragraph in section 3.1 to clearer explain and support the analysis of the differences between the RCMs and ERA5 and between the models themselves. In addition, we have changed figure 3 where we added the difference plots. Please see the discussion in the first general comment and refer to the revised text (lines 247-271) mentioned there.

**11 Line 264: How do the cyclones 'turn into precipitation when they reach the coast'? Do you mean that at the coast, orographic ascent causes water vapour to be converted into water droplets, which then grow into precipitation droplets?**

The reviewer is correct, we referred to the orographic mechanism that trigger convection over coastal areas. We have changed the paragraph to better explain the precipitation distribution associated to the winter extreme cyclones (L289-293) as follow:

"*This precipitation pattern is associated with winter cyclones generally coming from the west, as indicated by Flaounas et al. (2015) and Raveh-Rubin and Flaounas (2017) and interacting with the complex orography of the basin, increasing precipitation over coastal areas. The distribution of convective precipitation (Fig. 4e) is mainly concentrated over the sea, where the potential temperature lapse rate is low (i.e., low atmospheric*

*stability, Fig 4d), and close to the coastal regions where the sharp transition between sea and land fosters the convection processes*."

**12 Line 267: Can the authors expand on their statement that the transition between sea and land fosters the convection processes? Are you referring to convergence at the coast?**

The reviewer is correct, we were referring to the convergence and then convection processes in the coastal areas, especially in Italy and Balkans. Please refer to the comment 11 and the revised text (L289-293) mentioned there.

**13 Figure 4: How do these composites differ from the full winter climatology? Perhaps anomaly fields could be shown?**

Please see the discussion in the second general comment, where we have shown anomaly fields between cyclones-composite and climatological scale atmospheric fields (Fig. S2 in supplementary). This helps us to highlight the stronger surface fluxes, wind speed and precipitation during the cyclonic events and to emphasise the importance of the results.

**14 Fig 6: The order of the figures is different to that in figure 5 which confused me for a while.**

We have corrected both figure 5 and 6 in the revised manuscript.

**15 Fig 7: The order of the figures is different to those in fig 4. Could they be reordered to be consistent?**

We have corrected figure 4 in the revised manuscript which now shows the same atmospheric fields and follows the same order of figure 7.

**16 Figure 8: Since these fields are similar to those shown in previous figures, I do not think they add much to the analysis.**

We have changed figure 8 by showing only the temperature and specific humidity fields at 950 hPa and 850 hPa. These figures help us to prove that the SST differences have an impact throughout the entire PBL and not only at the surface. In fact, the vertical transport processes provide an increase of energy at different vertical level, destabilizing the PBL and making the atmosphere of the CPL warmer and moister at both 950 hPa and 850 hPa.

**17 Line 304: Reference is made to latent and sensible heat fluxes. Could these fields be shown instead of evaporation, wind speed, specific humidity and theta gradient? They are directly responsible for increasing the vertical exchange of heat and moisture between the Mediterranean Sea and overlying atmosphere so would be more relevant.**

The reviewer is correct, and we have added in the analysis the latent and sensible heat fluxes instead of the evaporation field.

**18 Figure 10: It appears that the MLD deepens more in the CPL model than in the CMEMS, why is this?**

We thank the reviewer to point out this interesting question. The reviewer is correct, the CPL model simulates a deeper MLD than the reanalysis. Likely, this depends on the different forcings coming from the atmospheric models (WRF in CPL and ERA5 in CMEMS) and intrinsic differences in the ocean models. However, further research would be needed to investigate the physical reason in detail, and this goes beyond the scope of the present paper.

**19 Figure 10: It is interesting that the MLD is twice as deep in winter than in autumn. Is this why the change in SST is so much smaller in winter?**

The reviewer is correct. In winter the mixed layer is much deeper and thus, the effect of the cyclones on ocean properties is weak, with a very low cooling of the temperature throughout the entire mixed layer depth. In autumn instead, the shallower mixed layer and the ocean stratification favour the upwelling processes caused by the strong winds during cyclones that enhance the surface moisture and heat releases in the atmosphere and, in turns, lowers the temperature of the ocean.

**20 Line 417: Is there evidence to support the statement that extreme cyclone significantly influences the Mediterranean climate? By climate, do the authors mean the long-term average conditions?**

There are several studies that demonstrate the strong influence of the cyclones on the Mediterranean climate. With climate we mean the long-term average conditions, the variability and the extremes. We talked about that in the introduction (L46-53) as follow:

"*Multiple studies indicate that cyclones in the Mediterranean region account for at least 70% of extreme rainfall events (Catto and Pfahl, 2013; Jansa et al., 2001; Nissen et al., 2013; Pfahl et al., 2014; Pfahl and Wernli, 2012), with deep convection and warm conveyor belt processes being the main contributors to heavy rainfall (Flaounas et al., 2018b, 2019). Additionally, these cyclones are responsible for the majority of extreme wind storms (Hewson and Neu, 2015; Nissen et al., 2010, 2014) and for the formation of high-impact weather events (Llasat et al., 2010, 2013). Those events produce a high variability in the evaporation and precipitation fields, playing a significant role in the Mediterranean Sea water budget (Flaounas et al., 2016; Romanski et al., 2012).*"

**Typographical errors**

**1. Line 20: 'Planet' should be 'planetary'.**

**2. Line 20: 'mixing of the turbulent processes' should be 'mixing by the turbulent processes'.**

**3. Line 98: 'insights on how' should be 'insights into how'.**

**4. Line 117: 'than STD' should be 'as STD'.**

**5. Line 174: 'planet boundary layer' should be 'planetary boundary layer'.**

**6. Line 179: What is the 'e' after STD? Is this a typographical error?**

**7. Line 226: 'upscaled at ERA5 resolution' should be 'upscaled to ERA5 resolution'**

**8. Line 290: 'norther' should be 'northern'.**

**9. Line 350: 'THETA' is represented as q elsewhere.**

**10. Line 53: 'Not statistically significant differences' should be 'statistically insignificant differences'.**

We have corrected all the typographical errors, thanks for pointing them out.

---

## Author Response (AR2)

**Reviewer #1:**

**The authors have addressed my comments adequately.**

We thank the Reviewer for dedicating time to review our manuscript and for the observations that have raised the quality of the paper.

Below, the Reviewer will find our detailed, point-by-point, answers.

**I have two minor suggestions:**

**Reconsider the phrasing of the following sentence (line 269): " In this context, atmosphere-ocean coupling is expected to have a stronger influence on the physical processes within the cyclone systems, and a rather weaker effect on their formation, distribution, and track characteristics." In my opinion, the differences in Figure 3d are not so small. In some locations, they are larger than 10. Considering that the largest number of cyclones in ERA5 is about 50, a difference of 10 means at least 20% and probably more (depending on the exact pixel). When plotted with the same color scale, it looks small because the difference is smaller in general than the difference relative to observations (panels 3b,3c). It would probably look clearer if you used the percentage instead of the total number.**

We thank the Reviewer for raising this point. The Reviewer is correct, when looking at the differences in cyclone distribution between CPL and STD (Fig. 3d), the atmosphere-ocean coupling appears to have an impact in the location of the cyclone minima over the Sea, and a rather weak effect only on the seasonal cycle and statistics of the tracks. Based on this, we have revised our conclusions in Section 3.1 as follow (L266-273):

"*In contrast, changes in the SST distribution primarily affect the location of cyclone minima over the sea (Fig. 3d), leading to differences between STD and CPL over the Ionian and Tyrrhenian Sea. Interestingly, when compared to ERA5, the CPL model reproduces the cyclone distribution over the sea slightly more accurately than STD, with a lower root mean square error (RMSE) in the location of cyclone minima (2.16 vs. 2.17 for STD), despite having greater degrees of freedom (i.e., the ocean domain in CPL is not constrained to observed SST). In conclusion, cyclone systems arise from a combination of large-scale processes (external to the cyclone) and small-scale processes (internal to the cyclone). In this context, atmosphere-ocean coupling is expected to have a stronger influence on the physical processes within the cyclone systems, and a minor, yet significant impact on their locations.*"

Following the Reviewer's suggestion, we have updated Figure 3 to display cyclone centre densities (CCD) as percentages, normalized by the total number of cyclones (the 500 most intense). The revised Figure 3 is shown below.

We have also changed the sentence in lines (251-252) to clarify the definition of the total number of CCD occurrences:

"*Consequently, the CCD maps (Fig. 3) indicate the number of cyclone occurrences at each grid point, normalised by the total number of cyclones (the 500 most intense).*"

[Figure]

**Figure 3: Cyclone centre densities (CCD) for the 500 most intense cyclones in ERA5 (a), along with CCD differences between STD and ERA5 (b), CPL and ERA5 (c), and CPL and STD (d). The values are normalised by the total number of cyclones (i.e. 500) and expressed as percentage. To highlight the cyclones' area of influence, each centre is represented by a circular area with radius of 1.5 degrees around the tracked minimum SLP point.**

**Also, I think that if a coupled model has a better representation of the cyclones (though it is minor, according to the calculation of RMSE of the distance between observed and simulated cyclone location at maximum intensity) is still impressive, considering it has much larger degrees of freedom (i.e., the ocean domain is not constrained to observed SST). I would say something about this in the manuscript.**

We thank the Reviewer for this suggestion. In Section 3.1, we have added the following sentence (L268-271):

"*Interestingly, when compared to ERA5, the CPL model reproduces the cyclone distribution over the sea slightly more accurately than STD, with a lower root mean square error (RMSE) in the location of cyclone minima (2.16 vs. 2.17 for STD), despite having greater degrees of freedom (i.e., the ocean domain in CPL is not constrained to observed SST).*"

**Reviewer #2:**

**Overall comment**

**I appreciate the authors' efforts toward the improvement of the manuscript. I did find the manuscript has been substantially improved.**

**Nevertheless, some of the concerns over the earlier version of the manuscript have not been fully resolved, and as a consequence, there are still a few issues to be addressed, as commented specifically below.**

**Therefore, I think that another major revision is required before the manuscript could be published.**

We thank the Reviewer for dedicating time to review our manuscript and for the observations that have raised the quality of the paper.

Below, the Reviewer will find our detailed, point-by-point, answers.

**Comments (In the following, line numbers correspond to those in the revised manuscript with a trace of modifications unless otherwise specified)**

**A) After I read the authors' response to my comments on the earlier version of the manuscript and the revised manuscript, I have better grasped the arguments of this study. Specifically, I understand that the SST difference between STD and CPL does not have a large impact on the cyclone density distribution. However, it leads to the impression that the main conclusions of this study are somewhat unclear. Probably I still do not fully figure out what the authors' are arguing, but it is recommended that the authors should make their standpoints and conclusions of this study clearer as readers easily can find a way to follow.**

We thank the Reviewer to point out this issue. We believe that the main findings and the novelty of our work was really improved in the revised manuscript. However, the valuable feedback raised by the Reviewer helped us to improve the clarity of the main findings presented in the conclusion (Section 4, L461-467):

"*This research demonstrates for the first time the ability of the coupled model to coherently simulate the entire atmosphere-ocean system, offering novel insights into how extreme Mediterranean cyclones influence both atmospheric and oceanic processes. Specifically, it investigates how energy released at the sea surface during these events affects the atmospheric boundary layer and the ocean mixed layer. Furthermore, comparing the models allows for quantifying the impact of sea surface available energy on precipitation and surface wind speed associated to extreme Mediterranean cyclones. These findings are of crucial importance in the climate change context, since atmosphere-ocean coupled RCMs give the possibility to reduce the uncertainty deriving from coarse-resolution SSTs coming from the global models.*"

**B) I am still puzzled by the interpretation of the SST differences between the observations and experiments. I expect that the SST difference between CPL and STD evaluates the effect of coupling in the model's domain.**

**For instance, Figs. 5b and 6b tell us how air-sea coupling in the Mediterranean Sea affects cyclone-related environments in the model. However, this study compares the climatological SST in STD and CPL with the MED-REP-L4 SST, instead of ERA5 SST, which is used as the boundary condition of STD. The climatological SST difference between STD and MED-REP-L4 (Figs. 5d and 6d) is seemingly due simply to the difference in data used, and why do the authors show it? In addition, I understand that the SST difference between CPL and MED-REP-L4 (Figs. 5c and 6c) corresponds to the bias of the model (as mentioned in Section 3.3), suggesting that the CPL model has a warm (cool) bias up to 1~2K in its climatological SST. This contrasts with the SST in STD, which is much closer to the observations (MED-REP-L4) even when ERA5 SST is prescribed.**

**Then Section 3.4 argues that the SST difference between STD and CPL is the dominant factor shaping both the sea surface fluxes and the precipitation and wind speed differences associated with the extreme cyclones. Does this indicate that, for example, "the analysis shows the ability of the coupled model to coherently represent the dynamic and thermodynamic processes associated with extreme cyclones across both the atmosphere and the ocean" (in the abstract)? I agree with this in terms of the ocean response to cyclones, but I am afraid that this is rather not valid for the processes in the atmosphere, given the SST bias in CPL. In addition, I understand that mechanisms behind the SST difference between STD and CPL, in other words, "coupling effect", are beyond the scope of the present study (as in the authors' reply). Nevertheless, the way in which the SST difference should be interpreted needs further**

**clarification. The authors might also want to mention that the mechanisms for the SST difference need an additional study in the manuscript.**

We appreciate the Reviewer's feedback on this point. To better highlight our findings, we have modified Figure 5 as follows:

- **Fig. 5a**: SST distribution in the MED-REP-L4 dataset.

- **Fig. 5b**: SST climatological differences between STD and MED-REP-L4.

- **Fig. 5c**: SST climatological differences between CPL and STD.

- **Fig. 5d**: SST differences between CPL and STD during extreme cyclone events.

The updated Figure 5 is provided below. We have applied the same modifications to Figure 6 (SST in SON).

[Figure]

**Figure 5: Map of SST from the MED-REP-L4 observational dataset in winter (a). Climatological winter SST differences between STD and MED-REP-L4 (b) and between CPL and STD (c). SST differences between CPL and STD during extreme winter cyclone events (d). The white colour indicates no significant differences at 5 % confidence level. Δ values represent the domain average of the differences where the values are statistically significant.**

In Section 3.3 (SST analysis), we focus on the SST differences between CPL and STD. Since the STD simulation uses prescribed SST from ERA5 reanalysis, we firstly validate ERA5 SST against the high-resolution observational dataset MED-REP-L4 (Fig. 5b). Then, to investigate the SST differences between CPL and STD, we compare them at both the climatological scale (Fig. 5c) and during the extreme cyclone events (Fig. 5d). The similarity between these differences (Fig. 5c vs. Fig. 5d) suggests that they are not directly influenced by cyclonic activity but rather reflect a climatological bias in the MITgcm ocean model embedded in CPL compared to ERA5. Similar results are found for SON (Fig. 6c vs. Fig 6d). These findings are presented concisely in the manuscript (L323-328) as follow,

"*During extreme winter cyclone events, compared to STD, the CPL model is remarkably warmer, up to 1.5 °C, over most of the Mediterranean Sea, except for the northern part of the Adriatic Sea and, to a smaller degree,*

*the Eastern Sea where the difference has opposite signs (Fig. 5d). SST differences are not associated with the occurrence of the cyclones, but rather to the climatological bias of explicitly resolved SST by the coupled model. Indeed, the same difference appears also when comparing the SST climatology in CPL with STD (Fig. 5c vs. Fig. 5d), while limited differences are found between STD and MED-REP-L4 (Fig. 5b).*"

The significant SST differences between the models motivates us to evaluate their impact on atmospheric processes during extreme cyclone events (section 3.4), when air-sea fluxes and convective processes are expected to be stronger compared to climatological values (as shown in figure S2). Finally, we examine the ocean model to investigate the ocean's response during the extreme cyclones. The findings from Sections 3.4 and 3.5 support our conclusion that the CPL model is able to coherently simulate the dynamic and thermodynamic processes associated with extreme cyclones in both the atmosphere and ocean.

In fact, in the CPL system, the atmospheric model responds to a warmer SST fostering surface latent and sensible heat fluxes, leading to modifications in atmospheric properties up to the top of the boundary layer, with impact on both 10 m wind speed and convective precipitation. At the same time, the strong winds during cyclones, enhancing evaporation and surface heat releases, lead to ocean cooling in the mixed layer and favour the turbulent mixing processes.

If the atmosphere and ocean were not coherently coupled, we would not observe significant differences between CPL and STD. Thus, the CPL model's response to a warmer SST is a direct consequence of the coupling system. However, the SST differences between CPL and STD (ERA5) stem from how the MITgcm ocean model simulates the Mediterranean Sea. Understanding these mechanisms requires further investigation and is beyond the scope of this paper. We have now clarified this point in the manuscript (L333-335):

"*Further information on the validation of the ocean system of the CPL can be found in Anav et al. (2024) across all seasons. However, the underlying mechanism responsible for CPL's climatological SST bias remains unclear and requires further investigation, which is beyond the scope of this study.*"

**C) The authors discuss the influence of atmosphere-ocean coupling on physical processes (i.e., diabatic heating related to precipitation) within cyclone systems based on regional model experiments. The discussion by and large makes sense to me. Nevertheless, I consider that the authors need to be careful when discussing "atmosphere-ocean coupling" using a regional model. Diabatic heating within cyclone systems is a moisture sink, which requires a moisture supply. Since moisture is supplied mostly by evaporation from the ocean, I would say that atmosphere-ocean coupling (in any place) is pivotal for diabatic heating within cyclone systems, in the first place. It should be noted that, in regional model experiments with relatively small domains (compared to, say, the entire North Pacific), only moisture supply from the ocean within the model's domain is included in the SST's influence. By contrast, moisture supply from the ocean outside the model's domain is included in the influence of lateral boundaries. Recent studies (Papritz et al. 2021; Okajima et al. 2024) show that moisture to precipitate within extratropical cyclones comes largely from remote regions (oceanic frontal zones or subtropics). Discussing the aspect of remote moisture supply for extratropical cyclones referring to those studies would further clarify the authors' arguments for the relative importance of (rather local) atmosphere-ocean coupling to large-scale forcings.**

The Reviewer is correct, the moisture during Mediterranean cyclones events comes not only from the Mediterranean Sea but also from the Atlantic, by advection processes. In our comparison, both models share the same lateral boundary conditions, meaning that the moisture entering the domain from external sources is identical in both simulations, as it is inherited from the ERA5 reanalysis. Therefore, our findings are based on the local air-sea fluxes within the Mediterranean, and their role on the atmospheric processes during extreme cyclone events.

We appreciate the Reviewer for sharing these relevant studies. In response, we have added a sentence in section 3.4 (L346-350) acknowledging the relative importance of both remote (external to the cyclone's area of

influence) and local (within the cyclone) moisture sources in contributing to precipitation associated with extratropical cyclones.

"*In the Mediterranean, precipitation within the cyclones is sustained both by moisture advected from remote regions, i.e. the Atlantic Ocean, as well as by local evaporation over the Mediterranean Sea (Flaounas et al., 2016; Raveh-Rubin and Wernli, 2016), similarly to what occurs in extratropical cyclones over open oceans (Okajima et al., 2024; Papritz et al., 2021). However, since CPL and STD share the same lateral boundary conditions from ERA5, the only difference in terms of moisture supply derive from their distinct interactions with the Mediterranean Sea surface*.*"*

**Other comments**

**1. L278: "This upper tropospheric forcing" -> "This large-scale upper tropospheric forcing" will be better.**

It has been corrected.

**2. L286-287: The argument "therefore, differences from ERA5 should not be taken purely as a weakness of RCMs, but rather as a result of differences when reproducing atmospheric processes" is un convincing and thus unnecessary. While I do not mean that this particular model is not useful for investigating Mediterranean cyclones, there is a difference. I consider it sufficient to describe differences between the model results and ERA5 and to suggest potential related factors.**

We thank the Reviewer for raising this point. We initially included this sentence in response to the previous Reviewers' comments on the comparison of the models with ERA5. However, we agree that it can be removed, and we have now corrected the text accordingly.

**3. Fig. 9: It would be better to show numbers with 2 (or 3 if needed) decimal points.**

We thank the Reviewer for this suggestion. We have updated Figure 9 (DJF) and Figure S6 (SON) to display values with two decimal points.

**4. L433, L437 (in the manuscript without a trace of modifications): "Click or tap here to enter text." -> error?**

We thank the Reviewer for highlighting this issue. It was an error, and we have now fixed it.

**5. Regarding the ocean response to extreme cyclones: Kuwano-Yoshida et al. (2017) pointed out that explosive cyclones over the North Pacific influence the ocean interior reaching 2000 m depth by inducing anomalous cooling and upward flow, based on a case study using an eddy-resolving OGCM. It would be a good comparison for the result of the present study.**

We thank the Reviewer for sharing these interesting studies. In Section 3.5, we have added a comment on the impact of cyclones over open oceans, which is significantly larger than over the Mediterranean but follows the same mechanism. Please refer to lines L420-421:

"*Interestingly, these mechanisms are similar to those over open oceans (Kuwano-Yoshida et al. 2017), although with lower magnitude*.*"*

**6. Unfortunately, there are still many expressions "storm track method", which the authors might want to replace with "cyclone tracking algorithm" in my comment on the previous version of the manuscript.**

Apologies for the misunderstanding regarding your previous comment (minor comment 20 from the first revision). We agree that "cyclone tracking" is more precise than "storm track," which has a broader meaning. We have now corrected this throughout the manuscript.

**References**

Kuwano-Yoshida, A., Sasaki, H., & Sasai, Y. (2017). Impact of explosive cyclones on the deep ocean in the North Pacific using an eddy-resolving ocean general circulation model. Geophysical research letters, 44(1), 320-329.

Okajima, S., Nakamura, H., & Spengler, T. (2024). Midlatitude oceanic fronts strengthen the hydrological cycle between cyclones and anticyclones. Geophysical Research Letters, 51(6), e2023GL106187.

Papritz, L., Aemisegger, F., & Wernli, H. (2021). Sources and transport pathways of precipitating waters in cold-season deep North Atlantic cyclones. Journal of the Atmospheric Sciences, 78(10), 3349-3368.